# The NF-κB Pharmacopeia: Novel Strategies to Subdue an Intractable Target

**DOI:** 10.3390/biomedicines10092233

**Published:** 2022-09-08

**Authors:** Daniela Verzella, Jessica Cornice, Paola Arboretto, Davide Vecchiotti, Mauro Di Vito Nolfi, Daria Capece, Francesca Zazzeroni, Guido Franzoso

**Affiliations:** 1Department of Biotechnological and Applied Clinical Sciences (DISCAB), University of L’Aquila, 67100 L’Aquila, Italy; 2Department of Immunology and Inflammation, Imperial College London, London W12 0NN, UK

**Keywords:** nuclear factor κB, NF-κB inhibitors, cancer therapy, targeted therapy

## Abstract

NF-κB transcription factors are major drivers of tumor initiation and progression. NF-κB signaling is constitutively activated by genetic alterations or environmental signals in many human cancers, where it contributes to almost all hallmarks of malignancy, including sustained proliferation, cell death resistance, tumor-promoting inflammation, metabolic reprogramming, tissue invasion, angiogenesis, and metastasis. As such, the NF-κB pathway is an attractive therapeutic target in a broad range of human cancers, as well as in numerous non-malignant diseases. Currently, however, there is no clinically useful NF-κB inhibitor to treat oncological patients, owing to the preclusive, on-target toxicities of systemic NF-κB blockade. In this review, we discuss the principal and most promising strategies being developed to circumvent the inherent limitations of conventional IκB kinase (IKK)/NF-κB-targeting drugs, focusing on new molecules that target upstream regulators or downstream effectors of oncogenic NF-κB signaling, as well as agents targeting individual NF-κB subunits.

## 1. Introduction

Since its discovery by Ranjan Sen and David Baltimore in 1986, the nuclear factor κB (NF-κB) transcription-factor pathway has been the focus of intense investigation, reflecting its ubiquitous nature and pivotal roles in essential physiological processes, such as the coordination of immune and inflammatory responses and cell survival, and the etiopathogenesis of a myriad of human diseases, including cancer [1,2,3,4,5,6]. In mammalian cells, the NF-κB family consists of five DNA-binding subunits, known as RelA (p65), RelB, c-Rel, NF-κB1 (p105/p50) and NF-κB2 (p100/p52), which assemble into almost all possible combinations of homodimeric and heterodimeric complexes, the most common and abundant being the NF-κB1/p50-RelA heterodimer [4,7,8,9,10]. All five subunits share a highly conserved amino-acid region, referred to as the Rel-homology domain (RHD), which mediates NF-κB dimerization, DNA binding, and nuclear translocation, as well as the association of NF-κB dimers with IκB regulatory proteins [4]. Normally, NF-κB complexes lay dormant in the cytoplasm of resting cells, where they are bound to inhibitory proteins of the IκB family. NF-κB is activated from these latent cytoplasmic pools in response to stimuli that induce the phosphorylation of IκBs by the IκB kinase (IKK) complex, leading to the sequential K48-linked polyubiquitination and proteolysis of the inhibitors by the SKP1-Cullin 1-F-box protein (SCF) E3 ubiquitin-protein ligase complex, SCF^βTrCP^, and 26S proteasome, respectively [8,10,11,12,13].

Several pathways of NF-κB activation have been characterized, including the canonical, non-canonical, and atypical pathways. In the canonical NF-κB pathway, the engagement of cognate receptors by inflammatory cytokines—such as tumor necrosis factor (TNF)α and interleukin 1β (IL-1β), microbial products, damage-associated molecular patterns (DAMPs), or antigens—initiates distinct signaling cascades that result in the activation of an IKK complex typically formed by the catalytic subunits, IKKα/IKK1 and IKKβ/IKK2, and the adaptor protein, IKKγ (also known as NEMO). Upon IKK activation, IKKβ phosphorylates IκBα or the other canonical IκB-family members, IκBβ and IκBɛ, via an IKKγ/NEMO-dependent mechanism, leading to the polyubiquitination and proteasomal degradation of IκB proteins [4,8,10,11,14]. The removal of the inhibitors enables the release of canonical, RelA- and/or c-Rel-containing NF-κB complexes, which then enter the nucleus, where they bind to distinctive DNA elements situated in the promoter or enhancer regions of their target genes, coordinating the expression of transcriptional programs that govern immune and inflammatory responses, cell survival, and other host defense mechanisms [4,7,8,10,13] (Figure 1).

In contrast to canonical NF-κB signaling, the non-canonical (or alternative) NF-κB pathway is activated by a distinct group of ligands of the TNF superfamily, such as CD40 ligand (CD40L), lymphotoxin β (LTβ), B-cell activating factor (BAFF), and receptor activator of NF-κB ligand (RANKL), and primarily regulates gene expression programs that control the development of secondary lymphoid organs, B-cell survival and differentiation, dendritic-cell activation, and bone morphogenesis [4,9,10,15,16]. Non-canonical NF-κB signaling is also independent of IKKβ and IKKγ/NEMO; rather, it relies on NF-κB-inducing kinase (NIK) for signal-induced phosphorylation of IKKα, which, in turn, phosphorylates NF-κB2/p100, leading to SCF^βTrCP^-dependent polyubiquitination and partial proteasome-mediated proteolysis of NF-κB2/p100 and the generation of mature NF-κB2/p52-RelB heterodimers. NF-κB2/p100 processing then enables NF-κB2/p52-RelB complexes to translocate to the nucleus and regulate the transcription of non-canonical NF-κB target genes [9,10,11,13] (Figure 2). The molecular details and biological functions of canonical and non-canonical NF-κB signaling and of other pathways of NF-κB activation, such as the atypical NF-κB activation pathway, triggered upon genotoxic stress, have been extensively discussed in previous reviews [17].

In addition to its important physiologic roles in host defense responses to microbial pathogens, injury, and stress, the IKK/NF-κB system has emerged as a major driver of etiopathogenesis in many, if not all, of the most serious threats to global human health, including atherosclerosis, autoimmunity, chronic inflammatory diseases, neurodegenerative conditions, type-2 diabetes (T2D) and other metabolic imbalances such as obesity, as well as cancer. In contrast to physiologic NF-κB activation, which is intrinsically rapid and self-limiting due to a multitude of feedback mechanisms that ensure the prompt cessation of the NF-κB response [9,10], pathologic NF-κB signaling is characteristically persistent and unrestrained due to lasting injury or genetic alterations that result in tissue damage, chronic inflammation, and oncogenesis. Accordingly, constitutive NF-κB activation is a hallmark of most human cancers, where it drives tumor progression, disease recurrence, metastatic spread, and therapy resistance by inducing transcriptional programs that sustain cancer cell survival and proliferation and fuel tumor-promoting inflammation [6,18,19]. A large body of evidence indicates that NF-κB also contributes to epithelial-to-mesenchymal transition (EMT), metabolic reprogramming, tissue invasion, angiogenesis, and avoidance of immune destruction, thereby impacting most aspects of cancer. In some malignancies, such as multiple myeloma (MM), diffuse large B-cell lymphoma (DLBCL), Hodgkin’s lymphoma (HL), and glioblastoma multiforme (GBM), NF-κB is often constitutively activated by recurrent genetic alterations that target core components or upstream regulators of both the canonical and non-canonical NF-κB pathways. In the majority of cases, however, constitutive NF-κB activation in cancer occurs as a result of mutations in conventional oncogenes or tumor suppressors, such as *RAS* and *PTEN*, respectively, or inflammatory stimuli and other cues that originate from the tumor microenvironment (TME). Many of these TME-derived signals are themselves dependent on NF-κB activation in non-malignant cells, thus fueling a positive feedback cycle that perpetuates cancer-cell survival, tumor-based inflammation, metastatic dissemination, and cancer immune evasion [6,7,20,21,22,23].

The existing body of genetic, biochemical, and clinical evidence provides a strong rationale for therapeutically targeting the NF-κB pathway in a wide range of human cancers. However, despite the efforts made by the pharmaceutical industry over the last three decades, no specific NF-κB inhibitor has been clinically approved, due to the dose-limiting, on-target toxicities of systemic NF-κB blockade [10,20,24]. Underscoring this effort, in 2006, just before IKK inhibitors could be tested in animal models and human trials, there were already no fewer than 750 drug candidates aimed at blocking pathologic NF-κB activation [3,24]. However, notwithstanding these efforts and the progress made in the meantime in terms of unravelling the complex molecular mechanisms regulating the pathways of NF-κB activation and their biological functions in health and disease, the output of actionable medicinal interventions for targeting the IKK/NF-κB system in a clinical setting has been scant. Thus, considering the paramount importance of the IKK/NF-κB pathway in oncogenesis and the current shortfall in clinically useful NF-κB inhibitors, there is an urgent need for novel therapeutic strategies to therapeutically block pathogenic NF-κB signaling in human disease.

Historically, the barrier to targeting the NF-κB pathway in oncological settings has been achieving cancer-cell specificity, due to the ubiquitous and pleiotropic physiological functions of NF-κB [10,20,24]. In this review, we discuss the main strategies being developed to therapeutically target the IKK/NF-κB pathway in oncological patients, highlighting some of the most promising approaches intended to inhibit oncogenic NF-κB signaling while mitigating the inherent limitations of conventional IKK/NF-κB-targeting agents. These approaches have produced a large diversity of chemical entities, including small molecules, antibodies, peptidomimetics, small interfering (si)RNAs, and derivatives of microbial products. Thus, for ease of exposition, we have organized emerging strategies and the most representative compound classes according to their mode of action and the level in the NF-κB pathway at which they operate (Figure 3, Figure 4 and Figure 5). Given the breadth of the clinical and basic research underpinning this drug-discovery effort, it is not possible to discuss in detail all relevant aspects of this research. As such, for further information on the pharmacology of novel IKK/NF-κB-targeting approaches, as well as for a broader overview of the NF-κB pathway itself and its regulation and functions in physiology and disease, particularly in the context of oncogenesis, we refer to previous reviews where these topics have been more extensively discussed.

## 2. Therapeutic Targeting of the NF-κB Pathway in Cancer

### 2.1. Agents Acting Upstream of IKK

From a clinical perspective, the multiplicity of stimuli that are capable of initiating distinct signaling pathways of IKK activation provide a clear opportunity for therapeutic interventions aimed at inhibiting pathway-specific mechanisms of disease that depend on downstream aberrant NF-κB activation (Figure 3). Below, we enumerate the main receptor-dependent mechanisms that have been targeted to block oncogenic NF-κB signaling.

#### 2.1.1. TNF Receptors (TNF-Rs)

Due to their key roles in the pathogenesis of chronic inflammatory diseases and several cancer types, members of the TNF receptor (TNF-R) superfamily and their ligands have been a fertile ground for drug research [25]. Most of these receptors are also strong inducers of NF-κB signaling and, as such, have been targeted to block pathogenic NF-κB activation in cancer patients. The TNFα-specific neutralizing antibody, infliximab, which was approved in 1998 for the treatment of Crohn’s disease, is the first ever medicinal agent used in clinical settings to target the TNF-R system [25]. Infliximab and several other TNFα inhibitors, such as etanercept, adalimumab, golimumab, tocilizumab, abatacept, and certolizumab pegol, are currently also approved for the treatment of ulcerative colitis, rheumatoid arthritis, ankylosing spondylitis, psoriasis, and psoriatic arthritis [26]. However, their dose-limiting toxicities and significant immunosuppressive activities have precluded their clinical development in oncology [25,26]. One exception is the human TNFα analogue, tasonermin (Beromun), which has found a niche indication as adjunct therapy to sarcoma surgery to avoid amputation and for treating unresectable soft-tissue sarcoma of the limbs [27]. More recently, small molecule TNFα inhibitors, such as UCB-6786, UCB-6876, and UCB-9260, were developed to broaden the clinical utility of TNFα inhibition by reducing the high cost, side effects, and immunogenicity of TNFα-targeting biologics such as anti-TNFα antibodies [28]. These small molecules were shown to disrupt the interaction of TNFα with its receptor, TNF-R1, through an allosteric mechanism that stabilizes naturally occurring, distorted TNFα conformers, reducing TNF-R1 signaling and NF-κB activation in vitro. UCB-9260 also demonstrated good bioavailability and therapeutic activity, i.e., comparable to those of TNFα-targeting biologics, following oral administration in mouse models of TNFα-dependent inflammation [28]. Its superior oral bioavailability and selectivity for TNFα compared to other TNF-family members make UCB-9260 an attractive candidate for clinical development in oncology, as well as chronic inflammatory diseases.

Other receptors of the TNF-R superfamily, such as CD30 (also referred to as TNFRSF8), have been targeted with greater success for therapeutic interventions in oncology. For instance, the toxin-conjugated chimeric antibody, brentuximab vedotin (Adcetris), which binds to CD30, has been approved for the treatment of HL and anaplastic large-cell lymphoma (ALCL), two malignancies that express high surface levels of CD30 (Table 1). Moreover, antagonists of the RANK-RANKL signaling axis, which plays an essential role in osteoclast development through a mechanism that depends on NF-κB activation, are being used to treat bone disorders and bone-associated malignant pathologies [29]. For instance, the neutralizing human monoclonal antibody, denosumab, which binds to RANKL has been approved by the FDA to prevent osteoclast-mediated bone destruction in patients with hypercalcemia of malignancy (HCM) which is refractory to bisphosphonate treatment or bone metastasis from solid tumors, as well as in MM patients with skeletal-related events (SREs) [30]. Other antibodies and nanobodies targeting the RANK/RANKL interaction, such as JMT-103, are currently in clinical trials for similar indications in patients with bone metastases from solid tumors (NCT04630522). No small molecule inhibitor of either RANK or RANKL is currently in use in a clinical setting [31].

**Table 1 biomedicines-10-02233-t001:** Selection of clinical inhibitors targeting NF-κB pathway.

Compound	Molecular Target	Cancer Type	Ongoing Clinical Trials	Phase	Recruitment Status	Other Information	Refs
**Upstream IKKs complex**							
Brentuximab (Vedotin)	CD30	HL; ALCL	NCT01657331	1–2	Completed	Combination with bendamustine is safe and effective. Could be used as second-line therapy	[32]
		ALK + ALCL	NCT02462538	1–2	Terminated		
		ASM; Mast Cell Leukemia; SM	NCT01807598	2	Completed	BV is not active as a single agent in CD30+ advSM	[33]
		HL; systemic ALCL	NCT02939014	2	Completed	Positive benefit–risk profile for patients with R/R cHL and sALCL, confirming it as a potential treatment option	[34]
		MMe	NCT03007030	2	Recruiting		
		PTCL; Paediatric HL	NCT02169505	2	Terminated	Treatment performed after Allogeneic and Haploidentical Stem Cell Transplantation in High Risk CD30+ Lymphoma	
		Relapsed HL	NCT01900496	1	Terminated	Combination therapy with Rituximab	
Idelalisib (Cal-101)	PI3K	CLL	NCT01539291	3	Terminated	Double-Blind extension study evaluating the efficacy and safety of different dose levels of single-agent Idelalisib	
		R/R HL	NCT01393106	2	Completed		[35]
		FL; SLL	NCT02258529	2	Terminated	Combination with rituximab in previously untreated adults with FL, SLL	
Copansilib (BAY 80-6946)	PI3K	DBLCL	NCT04433182	2	Recruiting	Combination regimen with rituximab-bendamustine	[36]
		DLCBL	NCT04263584	2	Recruiting	Combination with rituximab-CHOP in patients with untreated DLBCL	
		R/R MCL	NCT04939272	1–2	Recruiting	Combination with venetoclax	
		MZL	NCT03474744	2	Recruiting	Combination with rituximab	[37]
Ibrutinib (PCI-32765)	BTK	CLL	NCT02801578	2–3	Completed	After one cycle at the prescribed 420 mg/d dose, ibrutinib dose can be reduced in subsequent cycles without loss of biological activity	[38]
		High risk Smoldering MM	NCT02943473	2	Terminated		
		NSCLC	NCT02321540	1b-2	Completed		
		MCL	NCT02558816	1–2	Active	The combination of obinutuzumab, ibrutinib, and venetoclax is well tolerated and provides high response rates, including at the molecular level, in relapsed and untreated MCL patients	[39]
		Relapsed, Refractory, or High-Risk Untreated CLL; SLL; RS	NCT02420912	2	Completed	The combination of nivolumab and ibrutinib has clinical activity in pts with RT with a 43% response rate	[40]
		CLL	NCT02315768	1–2	Active	Combination with obinutuzumab	
		Advanced FL	NCT02451111	2	Active	Combination with rituximab	
		MCL	NCT02356458	1–2	Terminated	Combination therapy with bortezomib followed by ibrutinib maintenance therapy	
IMO-8400	TLR 7, 8, and 9	DLBCL	NCT02252146	1–2	Completed	Dose-escalation study	
		WM	NCT02092909	1–2	Terminated	Lack of efficacy	
		WM	NCT02363439	1–2	Completed	Lack of efficacy	
LCL-161	cIAPs	SCL; GM	NCT02649673	1	Terminated	Combination with topotecan	
		CRC; NSCLC; TNBC; RCC	NCT02890069	1	Completed	Combination with several agents with immunomodulatory activity	
		R/R MM	NCT01955434	2	Completed	Tested as single agent and in combination with cyclophosphamide	
		MM	NCT03111992	1	Completed	Combination with CJM112, and PDR001	
Birinapant (TL32711)	cIAPs	Solid Tumors	NCT02587962	1–2	Terminated	Combination with pembrolizumab	
		High grade serum carcinoma	NCT02756130	1–2	Withdrawn	Combination with platinum-based chemotherapy	
		R/R Solid Tumors	NCT04553692	1	Recruiting	Combination with IGM-8444, Venetoclax, Bevacizumab, FOLFIRI,	
		Refractory Solid Tumors or Lymphoma	NCT00993239	1	Completed		
**NF-κB Core pathway**							
Icaritin	IKKα	HCC	NCT03236636	3	Recruiting	Tested as single agent	[41]
		HCC	NCT03236649	3	Recruiting	Tested in PD-L1+ advanced HCC	[41]
Bortezomib	Proteasome	High-risk MM	NCT02308280	2	Active	Following nonmyeloablative allogeneic transplant	
		R/R ALL	NCT02535806	2	Terminated		
		AML	NCT01736943	2	Completed	Combination with doxil/lipodox	
		AML	NCT01534260	1–2	Completed		
		R/R Lymphoma	NCT02613598	1	Completed	Combination with ruxolitinib	
		MCL	NCT02356458	1–2	Terminated	Combination with ibrutinib	
		MM	NCT01241708	3	Active		
		MCL	NCT03016988	2	Unknown	Combination with fludarabine and cytarabine	
		Neuroblastoma	NCT02139397	1–2	Active	Combination with DFMO	
		MM	NCT02237261	2	Completed		
Carfizomib	Proteasome	MM	NCT02302495	2	Active		
		MM	NCT02572492	2	Active		
		NET	NCT02318784	2	Completed		
		R/R NHL	NCT02142530	1	Completed	Combination with belinostat	
		R/R NHL; R/R HL	NCT02867618	1–2	Terminated	Combination with TGR-1202	
		TCL	NCT01738594	1	Terminated	Tested as single agent or combination with romidepsin	
		R/R Solid Tumors or Leukemia	NCT02512926	1	Recruiting	Combination with cyclophosphamide and etoposide	
Ixazomib (MNL-9708)	Proteasome	GBM	NCT02630030	1	Completed	Orally administered ixazomib reaches brain tumor tissue. Therapeutic potential needs to be determined	[42]
		MM, Lymphoma	NCT02924272	2	Active		
		Solid Tumors	NCT02942095	1	Active	Combination with erlotinib	
		MM	NCT02312258	3	Active		
		MM	NCT02477215	1–2	Completed	Tested as single agent or combination with bendamustine	
		B cell Lymphoma	NCT02898259	1–2	Active	Combination with lenalidomide plus rituximab	
MLN4924 (Pevonedistat)	NAE	AML	NCT01814826	1	Completed	Combination with azacitidine	
		AML; MDS	NCT02782468	1	Completed	Tested as single agent and in combination with azacitidine	
		AML; MDS	NCT02610777	2	Completed		
		AML	NCT03009240	1	Active	Combination with decitabine	
		AML	NCT04090736	3	Recruiting	Tested as single agent and in combination with azacitidine	
		Solid Tumors	NCT03057366	1	Completed		[43]
Vorinostat	HDAC	Solid Tumors	NCT04308330	1	Recruiting	Combination with chemotherapy in R/R solid tumors	
Azacitidine	DNMT	Breast cancer	NCT04891068	2	Recruiting	To determine the effect of low dose azacitidine therapy on tumor infiltrating lymphocytes (TILs) in primary tumors	
		R/R Peripheral TCL	NCT05182957	2	Recruiting	Combination with lenalidomide and anti-PD-1 monoclonal antibody	
Decitabine	DNMT	Solid Tumors	NCT03875287	1	Recruiting	Combination with cedazuridine	
		NHL	NCT04697940	1–2	Recruiting	Decitabine-primed CAR-T-cells in B-cell malignancies	
Eltanexor (KPT-8602)	NE	R/R MM; metastatic CRC; metatstatic CRPC; HR-MDS	NCT02649790	1–2	Recruiting		[44]
Selinexor (KPT-330)	NE	Metastatic CRC	NCT04854434	2	Active	Tested as single agent and in combination with pembrolizumab	
		MM	NCT03110562	3	Active	A once-per-week regimen of selinexor, bortezomib, and dexamethasone is a novel, effective, and convenient	[45]
ZEN003694	BET	Metastatic CRPC	NCT04986423	2	Recruiting	Combination with enzalutamide	
BMS-986158	BET	Pediatric Cancer	NCT03936465	1	Recruiting		
**NF-κB target genes**							
DTP3	GADD45β/MKK7	MM	MR/V027581/1	1–2	Active		
Venetoclax (ABT-199)	BCL-2	WM	NCT02677324	2	Completed	Venetoclax is safe and highly active in patients WM	[46]
		MCL	NCT02471391	2	Active	Combination with ibrutinib	[47]
		MCL	NCT02558816	1–2	Active	Combination with ibrutinib and Obinutuzumabis well tolerated and highly active	[39]
		AML	NCT02203773	1	Terminated	Combination with decitabine or azacytidine	[48]
		NHL, DLBCL	NCT02055820	1–2	Completed	Combination of with R-/G-CHOP in NHL demonstrated manageable safety and promising efficacy. Established a dose regimen for venetoclax plus R-CHOP in DLBCL	[49,50]
		R/R DLBCL	NCT03136497	1	Active	Combination with ibrutinib and rituximab	
		CLL	NCT03128879	2	Recruiting	Combination with ibrutinib	
		R/R CLL	NCT02427451	1–2	Active	Combination with obinutuzumab and ibrutinib	[51]
Navitoclax (ABT-737)	BCL-2	Advanced or metastatic solid tumors	NCT02079740	1–2	Recruiting	Combination with trametinib	
		Advanced Myeloid Neoplasms	NCT05455294	1	Recruiting	Combination with decitabine, and venetoclax	
Bevacizumab	VEGF-A	HCC	NCT03434379	3	Active	Combination with atezolizumab	[52]
		RCC	NCT02420821	3	Completed	Combination with atezolizumab	[53]
Vandetanib (ZD6474)	VEGFR	Metastatic Papillary or Follicular Thyroid Cancer	NCT00537095	2	Active, not recruiting	Tested for patients with thyroid neoplasms which are failing or unsuitable for radioiodine therapy	[54]
Axitinib (AG-013738)	VEGFR	Metastatic RCC	NCT00920816	3	Completed	Tested as single agent compared to sorafenib	[55]
Siltuximab	IL6	Metastatic Pancreatic Cancer	NCT04191421	1–2	Recruiting	Combination with spartalizumab	
Tocilizumab	IL-R6	Melanoma; NSCLC	NCT04940299	2	Recruiting	Combination with ipilimumab and nivolumab	[56]

Abbreviations: Hodgkin Lymphoma = HL; Anaplastic Large Cell Lymphoma = ALCL; Anaplastic lymphoma kinase + Anaplastic Large Cell Lymphoma = ALK + ALCL; Aggressive Systemic Mastocytosis = ASM; Mantle Cell Lymphoma = MCL; Systemic Mastocytosis = SM; Malignant Mesothelioma = Mme; Peripheral T Cell Lymphoma = PTCL; Chronic Lymphocytic Leukemia = CLL; High Risk Smoldering Multiple Myeloma = High risk Smoldering MM; Non-Small Cell Lung Cancer = NSCLC; Small Cell Lung Cancer = SCLC; Small lymphocytic lymphoma = SLL; Richter’s Syndrome = RS; Advanced Follicular Lymphoma = advanced FL; Diffuse Large B Cell Lymphoma = DLBCL; Waldenstrom’s Macroglobulinemia = WM; Gynaecologic Malignancies = GM; Colorectal Cancer = CRC; Triple Negative Breast Cancer = TNBC; Renal Cell Carcinoma = RCC; Relapsed/Refractory Multiple Myeloma = R/R MM; Multiple Myeloma = MM; Solid Tumors = ST; Acute Myeloid Leukemia = AML; Relapsed/refractory Acute Myeloid Leukemia = R/R ALL; Relapsed/refractory non-Hodgkin lymphoma = R/R NHL; Non-Hodgkin lymphoma = NHL; Relapsed/refractory Hodgkin lymphoma = R/R HL; T-Cell Lymphoma = TCL; Glioblastoma = GBM; Neuroendocrine Cancer = NET; Myelodysplastic syndrome = MDS; high risk-myelodysplastic syndrome = high risk-MDs; Relapsed/Refractory Chronic Lymphocytic Leukemia = R/R CLL; Hepatocellular carcinoma = HCC; Metastatic castration resistant prostate cancer = metastatic CRPC; Phosphoinositide 3-kinase = PI3K; Bruton’s tyrosine kinase = BTK; Toll-like receptor = TLR; Cellular inhibitor of Apoptosis Proteins = c-IAPs; NEDD8-activating enzyme = NAE; Histone deacetylase = HDAC; DNA methyltransferase = DNMT; Nuclear export = NE; Extra-terminal domain = BET; Growth arrest and DNA damage inducible beta = GADD45B; Mitogen-activated protein kinase kinase 7 = MKK7; B-cell lymphoma 2 = BCL-2; Interleukin-6 = (IL-6); Vascular endothelial growth factor-A = VEGF-A; Vascular endothelial growth factor receptor = VEGFR.

#### 2.1.2. Toll-Like Receptors (TLRs)

Toll-like receptors (TLRs) sense pathogen-associated molecular patterns (PAMPs) to initiate innate immune responses against infections. Except for TLR3, NF-κB activation by TLRs depends on the adaptor protein, MYD88, which is frequently targeted by gain-of-function mutations in hematological malignancies, such as DLBCL of the activated B-cell (ABC) subtype (ABC-DLBCL) and Waldenström’s macroglobulinemia (WM), where the oncogenic *MYD88^L265P^* mutation was shown to occur in over 30% and 90% of case, respectively [57]. As such, signaling intermediates of activated TLRs are attractive therapeutic targets in cancer patients, with several compounds currently in clinical use or in human trials for indications in oncology and autoimmune diseases. The synthetic oligonucleotide-based antagonist, IMO-8400, which targets endolysosomal TLR7, TLR8, and TLR9, was demonstrated to be effective in xenograft models of WM and DLBCL harboring *MYD88* mutations, but it is not clear why this inhibitor is more effective in cell harboring *MYD88^L265P^* compared to those lacking *MYD88* mutation. This TLR7,8,9 antagonist received orphan drug designation by the FDA in 2015 for the treatment of DLBCL [58]. However, subsequent clinical trials of IMO-8400 in patients with relapsed or refractory DLBCL and WM that carried gain-of-function *MYD88^L265P^* mutations were suspended for lack of efficacy (NCT02252146; NCT02092909; NCT02363439). Trials evaluating additional antisense oligonucleotides (such as IMO-3100 and IMO-9200) or small-molecule antagonists (such as CPG-52364) also targeting TLR7/TLR8/TLR9 in other indications have been completed, but the results have not been reported yet (NCT01622348; NCT00547014) [59]. Various agents targeting TLR2 or TLR5, such as CBLB612 (a synthetic lipopeptide agonist of TLR5), ISA-201B (a peptide agonist of TLR2), and OPN-305 (a monoclonal antibody inhibiting TLR2), also known as tomaralimab, have also been tested in phase I and II trials in patients with several cancer types, but the results have been published to date (NCT02778763; NCT03669718; NCT02363491). Additionally, the TLR3 agonist, poly-ICLC, is being investigated in human trials as an adjuvant cancer therapy (NCT04544007). Similarly, recombinant flagellin, which stimulates TLR5, is also being clinically evaluated as an adjuvant therapy in various indications, including cancer [60]. However, the most promising therapeutic strategy aimed at blocking oncogenic TLR signaling, particularly in tumors with *MYD88* mutations, is the pharmacologic inhibition of the downstream protein kinases, Interleukin-1 receptor-associated kinase (IRAK)1 and IRAK4. IRAKs are adaptor proteins which are recruited, along with MYD88, to TLR/IL1R receptors upon ligand binding. These proteins can, in turn, recruit TRAF6, which then triggers IKK activation [10,61,62,63]. The strategy of targeting IRAK proteins has demonstrated a clear efficacy against non-Hodgkin’s lymphomas (NHLs), myelodysplastic syndrome (MDS), T-cell acute lymphoblastic leukemia (T-ALL), and melanoma in preclinical settings [64,65], while small-molecule IRAK1/4 inhibitors, R835 and PF-06650833, are currently in clinical development for use in autoimmune and chronic inflammatory diseases, such as rheumatoid arthritis (RA) and systemic lupus erythematosus (SLE). The small-molecule IRAK1 inhibitor, pacritinib, was recently approved by the FDA as an anti-cancer medication to treat myelofibrosis. Moreover, the orally available small-molecule IRAK4 inhibitor, CA-4948 (emavusertib), is currently in phase I trials in NHL, MDS, and acute myeloid leukemia (AML), as it demonstrated anti-cancer efficacy signals and a favorable safety profile in preliminary clinical data (NCT04278768; NCT03328078; NCT05178342).

#### 2.1.3. Cellular Inhibitor of Apoptosis Proteins (c-IAPs)

Cellular Inhibitor of Apoptosis Proteins (c-IAP) play an important role in the regulation of both canonical and non-canonical NF-κB signaling. Following TNF-R1 engagement by TNFα, receptor-bound c-IAP1/2 contributes to the recruitment of linear ubiquitination assembly complex (LUBAC), which activates NF-κB by catalyzing the ligation of linear ubiquitin to IKKγ/NEMO and RIP1 [66]. In contrast, in the non-canonical NF-κB pathway, the ubiquitin ligase activity of c-IAP1/2 promotes constitutive NIK degradation, thereby preventing non-canonical NF-κB activation [9]. Given these dual functions of c-IAP proteins in canonical and non-canonical NF-κB signaling, *c-IAP* genes are targeted by both gain-of-function and loss-of-function mutations in cancer [66]. The endogenous c-IAP inhibitor, second mitochondria-derived activator of caspases (SMAC), which binds to c-IAP proteins via its AVPI tetrapeptide motif, has provided a template for the structure-based design of small-molecule inhibitors of c-IAPs [67]. ASTX660 is one of non-peptidomimetic small-molecule inhibitors of c-IAP1/2 and X-linked inhibitor of apoptosis protein (XIAP); it is currently in phase I/II trials in patients with advanced-stage solid tumors and NHLs and is also in a phase II study in patients with peripheral and cutaneous T-cell lymphoma (TCL) (NCT05082259; NCT02503423; NCT04155580; NCT04362007; NCT05403450). Furthermore, the FDA recently granted breakthrough therapy designation to the orally available SMAC mimetic c-IAP antagonist, Debio 1143 (xevinapant), as a chemio- and radio-sensitizer in the front-line therapy of patients with previously untreated, unresectable head and neck squamous cell carcinoma (HNSCC), in combination with cisplatin-based concurrent chemoradiotherapy (CRT) [68,69]. Debio 1143 is also in phase II trials in other oncological indications and was granted orphan drug designation by FDA for the treatment of ovarian cancer (NCT04122625; NCT03871959; NCT02022098) [70]. APG-1387 is another SMAC mimetic and c-IAP antagonist that is currently in phase I/II trials for the treatment of advanced pancreatic carcinoma in combination with chemotherapy, with promising initial efficacy signals and overall good tolerability (NCT04284488; NCT04643405). In preclinical models, it has shown anti-tumor efficacy against HBV-positive hepatocellular carcinoma (HCC), ovarian cancer (OC), and nasopharyngeal carcinoma, either as monotherapy or in combination with other agents [71,72,73]. At least two other small-molecule SMAC mimetics have entered phase II trials in oncology: LCL161, an orally available compound which has been tested as a monotherapy and in combination with chemotherapy in patients with high-grade serous ovarian carcinoma (HGSOC), breast carcinoma, HNSCC, and relapsed or refractory MM; and birinapant (TL32711), which was tested in combination with keytruda (pembrolizumab) in patients with microsatellite stable (MSS) colorectal carcinoma (CRC) (NCT01955434; NCT01240655; NCT01617668; NCT02890069; NCT04553692; NCT02587962; NCT01486784; NCT00993239). Although both compounds have demonstrated some efficacy in clinical trials, and LCL161 had been shown to induce a dramatic tumor regression in xenograft models of HNSCC in combination with radiotherapy [74], they also exhibited severe dose-limiting toxicities, including cytokine-release syndrome [74]. 

#### 2.1.4. The Phosphoinositide 3-Kinase (PI3K)/AKT Pathway

The pathway mediated by Phosphoinositide 3-Kinase (PI3K) and the serine threonine kinase, AKT, also known as protein kinase B (PKB), is one of the most commonly deregulated signaling pathways in human cancers [75]. Here, PI3K/AKT signaling can be constitutively activated by several genetic mechanisms, including gain-of-function driver mutations in *PIK3CA*, the gene encoding the catalytic subunit, p110α; loss-of-function mutations or deletions in the tumor suppressor, *PTEN*; amplifications or gain-of-function mutations in genes encoding receptor tyrosine kinases (RTKs); and amplifications or gain-of function missense mutations in genes encoding one of the three AKT isoforms [76,77]. Upon activation by gene mutation or physiologic receptor stimulation, PI3K recruits AKT, which, in turn, activates its downstream effectors, mTOR and NF-κB, via direct phosphorylation of several mTOR- and NF-κB-pathway components [78], thereby promoting cell survival, proliferation, and anabolic metabolism [79,80]. As such, the PI3K/AKT signaling axis is a sought-after therapeutic target to block oncogenic signaling and downstream NF-κB activation in different cancer types. Several classes of PI3K inhibitors have been developed, including isoform-specific or dual PI3K inhibitors, pan-PI3K inhibitors, and dual PI3K/mTOR inhibitors, and are currently approved or in different stages of clinical development for the treatment of both solid and hematological malignancies [76]. In 2014, the orally available PI3Kδ inhibitor, CAL-101 (idelalisib), became the first PI3K-targeting agent to receive FDA approval for the treatment of relapsed or refractory chronic lymphocytic leukemia (CLL) in combination with the anti-CD20 antibody, rituximab, and as a monotherapy for the treatment of relapsed small lymphocytic lymphoma (SLL) and follicular lymphoma (FL), after at least two lines of prior therapy [81,82]. This was followed by the approval in 2017 of BAY 80-6946 (copanlisib), a pan-class I PI3K inhibitor with preferential activity against p110α and p110δ, for the treatment of relapsed FL following at least two lines of therapy [83]. Copanlisib is currently in phase II trials for endometrial cancer, cholangiocarcinoma, and NHL, including DLBCL and marginal zone lymphoma (MZL), and in phase III trials either as monotherapy or in combination with rituximab and chemotherapy for rituximab-refractory or relapsed indolent NHL (NCT04433182; NCT04263584; NCT04572763; NCT04939272; NCT03877055; NCT03474744). However, copanlisib requires intravenous administration and has been shown to cause severe adverse effects. As such, it has yet to be approved outside the United States (US) for medicinal use. In contrast, the orally available selective PI3Kα inhibitor, BYL719 (Alpelisib), is indicated in both Europe and the US in combination with hormonal therapy (fulvestrant) for the treatment of hormone receptor (HR)-positive, HER2/neu-negative advanced or metastatic breast cancer with *PIK3CA* gain-of-function mutations [84]. Similarly, the oral dual PI3Kδ and PI3Kγ inhibitor, IPI-145 (duvelisib), has been granted marketing authorization in both territories for medical use in adults with relapsed or refractory CLL or SLL, following at least two liners of prior therapy [85]. Duvelisib also received approval for the treatment of adults with relapsed or refractory FL after at least two lines of prior therapy but was recently pulled from the US market by the FDA because its post-marketing requirements are no longer necessary [85]. Numerous other PI3K inhibitors, including the PI3Kδ inhibitor, INCB050465 (parsaclisib), the dual pan-class I PI3K and mTOR inhibitor, GDC-0084 (paxalisib), and the pan-class I PI3K inhibitor, GDC-0941 (pictilisib), are currently in several phase II and phase III trials, generally in combination with other agents, for a wide range of solid and hematological cancers, including HNSCC, GBM, *PIK3CA*-mutated breast cancer, renal cell carcinoma (RCC), DLBCL, and mantle cell lymphoma (MCL) (NCT04434937; NCT04774068; NCT03765983; NCT03970447; NCT00996892). Although improvements in patient stratification strategies and the development of rational drug combinations are rapidly extending the clinical utility of PI3K inhibitors in the treatment of cancer patients, managing adverse effects, such as infections and inflammation, as well as drug resistance, remains a challenge [76,86]. Several AKT inhibitors acting via allosteric mechanisms, such as ARQ092 (miransertib), BAY1125976, MK-2206, and TAS-117, or via ATP-competitive binding, such as AZD5363 (capivasertib) and GDC0068 (ipatasertib), are currently in clinical trials for the treatment of solid tumors, either as monotherapies or in combination with other agents (NCT04980872; NCT01147211; NCT04770246; NCT03772561; NCT04464174). To date, however, no AKT inhibitor has been clinically approved, and most of them have thus far demonstrated limited clinical utility as single agents. Several strategies are therefore being investigated to improve the efficacy of these agents by combining them with other anti-cancer therapies, such as PD-1/PD-L1 immune checkpoint inhibitors, chemotherapeutic agents, or targeted therapies, such as the PARP inhibitor, olaparib, and CDK4/6 inhibitors [87].

#### 2.1.5. B Cell Receptor (BCR) Signaling

Constitutive IKK/NF-κB activation is the hallmark of many B-cell malignancies, such as HL, MM, DLBCL, and MCL [18,88]. In most of these cancers, NF-κB is frequently activated by recurrent genetic mutations targeting core components of the B Cell Receptor (BCR) complex, such as *CD79A* and *CD79B*, or upstream regulators of the canonical NF-κB pathway, such as *MYD88*, *CARD11*, and *TNFAIP3/A20* [89,90]. A separate group of mutations includes *REL* amplifications, loss-of-function mutations in *NFKBIA*, encoding IκBα, and gene alterations targeting components of the non-canonical NF-κB pathway, such as *NIK*, *TRAF2*, and *TRAF3* [89,90]. In some cases, such as in a subset of ABC-DLBCLs, NF-κB appears to be constitutively activated by non-genetic mechanisms that drive chronic BCR signaling and downstream assembly of the PKCβ and CARD11-BCL10-MALT1 (CBM) signaling complexes [89]. Consequently, in these tumors, malignant B cells frequently depend on constitutive NF-κB activation for survival and undergo apoptosis upon IKK/NF-κB inhibition [89,90]. Thus, there is a strong rationale for therapeutically targeting NF-κB in a broad range of B-cell cancers. Bruton’s tyrosine kinase (BTK) plays an essential role in driving NF-κB activation and B-cell survival downstream of the BCR, and, as such, has been targeted for therapeutic intervention in several hematological malignancies [91]. The irreversible oral BTK inhibitor, ibrutinib, is the first agent in this class to be approved for medicinal use by the FDA for the treatment of refractory MCL (2013) and, as the front-line therapy, in patients with newly diagnosed CLL (2014) and WM (2015) [92]. Ibrutinib is also indicated for the treatment of SLL and is under clinical investigation for in patients with FL, ABC-DLBCL, and B-ALL, as well as breast cancer as a potential immunotherapy [44,93,94]. In a phase I/II trial in patients with relapsed or refractory DLBCL, ibrutinib monotherapy produced an objective clinical response (OCR) of varying degree and duration in 37% of ABC-DLBCL cases, with higher response rates, reaching approximately 80%, in tumors with concurrent *CD79B* and *MYD88^L265P^* mutations [44]. Conversely, ibrutinib was largely ineffective in patients with germinal-center B-cell-like (GCB)-DLBCL tumors and the larger subset of patients with ABC-DLBCLs that rely on BCR-independent routes of NF-κB activation, such as *CARD11* gain-of-function mutations, *TNFAIP3/A20* loss-of-function mutations, or *MYD88*-only mutations [44]. A subsequent study by the same laboratory demonstrated that the sensitivity of a subset of ABC tumors to ibrutinib depends on the BTK-dependent assembly of a supercomplex formed by the MYD88, TLR9, and BCR (My-T-BCR) signalosomes, which co-localize with the mTOR signaling complex on endo-lysosomes to coordinate survival signaling by the NF-κB and mTOR pathways [95]. A recent phase III trial investigating the efficacy of ibrutinib in combination with rituximab, cyclophosphamide, doxorubicin, vincristine, and prednisone (R-CHOP) therapy in patients with newly diagnosed non-GCB DLBCL demonstrated a clear clinical benefit from ibrutinib treatment in a distinct subset of patients [44,93]. However, while ibrutinib is generally tolerated, with typically transient adverse effects, the relatively short duration of response—due to the onset of drug resistance and the preponderance of DLBCL cases that are naturally resistant to BTK inhibition, owing to their reliance on BCR-independent mechanisms of NF-κB activation—significantly limits the clinical utility of ibrutinib for managing patients with DLBCL [44,91,93]. Second-generation BTK inhibitors, such as acalabrutinib and zanubrutinib, have been approved by the EMA and the FDA as monotherapies or in combination with other agents for the treatment of newly diagnosed CLL or SLL and the treatment of patients with relapsed or refractory MCL [96]. Both agents are currently in clinical trials for use in additional oncological indications, as are several other BTK inhibitors, including orelabrutinib, tirabrutinib, rilzabrutinib, ABBV-105, and SAR-442168, as single agents or as part of polytherapies (NCT05189197; NCT05334238; NCT04947319; NCT03740529; NCT04562766; NCT03978520; NCT04458051; NCT02558816). While these newer agents and the development of combination therapies and improved patient stratification strategies may help to mitigate the onset of secondary drug resistance to BTK inhibition, there is an urgent need for novel strategies to therapeutically target oncogenic NF-κB signaling in NF-κB-addicted but BCR-independent B-cell tumors, including the majority of DLBCLs, which are inherently resistant to BTK inhibition.

Thus, despite the clinical results obtained by targeting upstream NF-κB signaling mechanisms in the treatment of many B-cell tumors, the clinical benefits of this strategy have been limited by dose-limiting toxicities, the primary resistance of certain cancer types, and the relatively early onset of secondary drug resistance. Thus, devising alternative NF-κB-targeting approaches, potentially including the identification of actionable protein–protein interactions involved in downstream IKK activation, as well improving the selection of effective drug combinations and the precision of diagnostic assays for patient stratification, may help to enhance the healthcare benefits to patients [10].

### 2.2. Agents Targeting Core Components of NF-κB Pathway

Targeting the principal players of the NF-κB signaling pathway has been the focus of academia and pharma for the last 30 years. This enormous research effort has led to the development of several therapeutic approaches which are able to modulate the NF-κB core pathway, the most important of which are described below (Figure 4) (Table 1). 

#### 2.2.1. IKK Complex

The IKK family is well known for its role in the activation of the NF-κB pathway during inflammation, immune cell activation, and tumorigenesis [4]. Aberrant activation of IKK family kinases is involved in different pathologic conditions including cancer, metabolic syndromes, and pathogen-associated diseases [97]. Therefore, the development of specific IKK inhibitors has been pursued by both academia and pharma since the discovery of the IKK complex. Although IKK inhibitors were proven to be effective at inhibiting the NF-κB pathway, when administered to experimental animals and human volunteers, they were found to be associated to severe toxicities due to the blockade of NF-κB ubiquitous functions [98]. 

*IKK**α/IKK**β inhibitors*. Although the goal has been to develop specific IKKβ inhibitors, these compounds also block IKKα due to the amino acid sequence similarities between the two kinases. There are three classes of IKKα/IKKβ inhibitors based on their mode of action: ATP analogues (i.e., SPC-839), allosteric modulators (i.e., BMS-345541), and agents interfering with the kinase activation loops (i.e., Thiol-reactive compounds) [3,10,99], as recently reviewed elsewhere [10]. However, no IKKα/IKKβ inhibitor has been clinically approved to date. The SAR-113945 selective anti-IKKβ drug, developed for the treatment of osteoarthritis, showed promise in phase I studies but failed to be proven effective in a larger phase IIb proof-of-concept study [100]. Other IKKβ inhibitors, like AS-602868 and CHS-828, were tested in hematological or solid cancers in phase I and II clinical trials, respectively, but these studies were discontinued due to dose limiting toxicities associated with the treatment and a lack of significant tumor response [101,102]. Similarly, MLN-0415 showed an unfavorable safety profile in a phase 1 trial when tested in inflammatory diseases [101]. VGX-1027 and IMD-1041 were tested in phase I and II trials for RA and chronic obstructive pulmonary disease, respectively, but they were not progressed for further development, although the reasons for this are still unknown, as no results have been published to date (NCT00883584; NCT00627120; NCT00760396). Teglarinad chloride was tested in phase I studies for several cancer types, but these trials were terminated prematurely due to financial constraints (NCT00724841; NCT00457574). Encouragingly, a prenylflavonoid derivative, Icaritin, previously identified as a PD-L1 regulator, has been demonstrated to be an IKKα inhibitor; it is currently being investigated in phase III trials as a single agent in China (NCT03236636 and NCT03236649) for the treatment of advanced HCC [41]. Mutagenesis assays revealed that Icaritin binds to IKKα at C46 and C178 residues and inhibits the NF-κB signaling pathway by blocking IKK complex formation [41]. The determination of the crystal structure of the IKKβ/inhibitor complex and structure-based in silico studies have provided a deeper understanding of IKKβ binding site conformation, thus paving the way for the discovery of novel IKKβ inhibitors [103]. Molecular docking analyses suggested that Yakuchinone A, a curcumin analog, inhibits IKK activity by forming multiple interactions within the catalytic site of IKK (amino acids 16–307) [104]. However, further investigations are required to prove the activity of this curcumin analog [104]. Similarly, the screening of 2000 bioactive compounds against IKKβ using an in silico chip-based assay system identified four new anti IKKβ agents (i.e., aurintricarboxylic acid, diosmin, ellagic acid, and hematein) which need to be experimentally tested to be recognized as new drug candidates [105]. Another class of IKK inhibitors is represented by those targeting the NEMO/IKKβ complex. Several NEMO binding domain (NBD) peptides which are able to block the interaction between NEMO and the IKK complex have been developed but have not yet been applied in a clinical setting due to their poor bioavailability and the low plasma stability observed both in vitro and in vivo [106]. Other potent small molecule inhibitors were developed in silico but failed to disrupt NEMO/IKKβ interaction in vivo. Recently, Yu and colleagues identified Shikonin (SHK), a natural product, as a potential NEMO/IKKβ complex inhibitor which is able to block the complex formation as well as the dissociation of the pre-formed complex, both in vitro and in vivo. Furthermore, SHK was shown to inhibit the expression of inflammatory proteins such as cyclooxygenase-2 (COX-2), inducible nitric oxide synthase (iNOS), Interleukin-6 (IL-6), and TNFα and revert the malignant phenotype by suppressing cancer cell proliferation via NF-κB blockade in both CRC cells and mouse models [107]. 

*TANK-binding kinase 1 (TBK1) and I**κB kinase**ε (IKK**ε) inhibitors.* The innate immune response mediated by TLRs and MYD88 involves IκB kinase ε (IKKε or IKBKE) and TANK-binding kinase 1 (TBK1), which share high sequence identity in the kinase domain [108]. IKKε is induced in response to viral/bacterial stimuli and cytokines and is expressed exclusively in thymus, pancreas, spleen, and peripheral blood leukocytes [109]. TBK1 is constitutively expressed and, along with IKKε, controls the activation of interferon regulatory factors (IRF)3, IRF5, and IRF7 but also mediates RELA phosphorylation [99,110]. The activation of the TBK1/IKKε/IRFs pathway occurs when DNA is aberrantly localized in the cytosol due to (1) the presence of pathogen-derived DNA; (2) self-DNA leakage from the nucleus following DNA damage; or (3) DNA release from mitochondria upon oxidative stress. This activation, in turn, triggers NF-κB-mediated pro-inflammatory cytokine production during immune response to pathogens and tumors [111]. The deregulation of TBK1 activity has been associated with a wide range of human diseases including autoimmune, inflammatory, and malignant disorders [112,113,114,115]. Similarly, IKKε is overexpressed in breast cancer and OC, as well as in RA, where a recent observational study (NCT02689115) showed a higher expression of IKKε in patients with active disease than in those in clinical remission [113,114,115].

The role of IKKε and TBK1 in the pathogenesis of human diseases has led to numerous research efforts to develop specific inhibitors. Although several small molecules showed efficacy in inhibiting IKKε and TBK1, no specific molecules targeting these kinases have been evaluated in clinical trials [116]. The first TBK1 inhibitor, BX795, and MRT67307 lacked specificity and were shown to target several protein kinases [117]. In the last few years, some drugs used in the clinic with unknown mechanisms were shown to target TBK1/IKKε. Amlexanox, a drug previously approved for the treatment of asthma and allergic rhinitis, was reported to target both IKKε and TBK1 [118]. A study of the crystallographic structure of Amlexanox showed that the molecule binds to TBK1 in the aminopyridine fragment of the catalytic domain [119]. This new knowledge led to the generation of an analogue series, bearing modification at the C7 position and with improved efficacy against TBK1 [120]. In the last five years, multiple chemical series have been published [116]. Of interest, GSK8612 is highly selective for TBK1 over IKKε and could be useful to dissect the biology of TBK1. The high selectivity of GSK8612 for TBK1 was demonstrated in Ramos cells, where this small molecule inhibited TLR3-mediated IRF3 phosphorylation. In human peripheral blood mononuclear cells (PBMCs) and THP1 cells, GSK8612 was shown to be effective at micromolar concentrations and to cause the abrogation of Interferon (IFN)α and IFNβ secretion [117]. However, further in vivo investigations are required to translate this inhibitor to a clinical setting. An emerging, powerful class of therapeutic agents is Proteolysis targeting chimeras (PROTACs), which induce the degradation of proteins of interest via UPS upon interaction with E3 ubiquitin ligases [121]. The selective degradation of TBK1 using PROTACs has been explored, leading to the development of Arvinas, which was subsequently optimized, yielding an improved compound which is able to efficiently degrade only TBK1 and not IKKε in vitro [122]. 

*NIK inhibitors*. As non-canonical signaling is frequently constitutively activated in several cancers, including MM and DLBCL, by recurrent genetic alterations, including *NIK* amplifications and *c-IAP1/2* and *TRAF3* deletions, NIK has emerged as an attractive target for therapeutic interventions, given its key role in controlling IKKα phosphorylation and NF-κB2/p100 processing [10]. Over the past decade, many NIK inhibitors (i.e., staurosporine, ZINC-1601221, B022, XT2, NIK SMI1, Cpd33, HTS Hit, 7H-pyrrolo [2,3-d]pyrimidin-4-amine-containing compounds, 4H-isoquinoline-1,3-dione, N-Acetyl-3-aminopyrazoles, Mangiferin, PDB:6Z1T, and PDB-6Z1Q) have been investigated, allowing researchers to identify the key residues of NIK involved in binding with inhibitors (Figure 4) [123,124]. Some of these molecules were tested in vivo and showed good efficacy in treating inflammatory, metabolic, and immune diseases [125,126,127,128]. The first NIK inhibitor that showed anti-cancer activity in several mouse models was TRC694. TRC694 has been demonstrated to suppress proliferation and tumor growth in MM cancer cells and xenograft models via inhibition of the p100 to p52 processing [129]. However, despite these encouraging preclinical results, no clear clinical candidates have emerged thus far.

#### 2.2.2. Ubiquitin and Proteasome Pathway

The ubiquitin-proteasome system (UPS) is one of the main mechanisms that regulates cellular protein levels and activity; it is responsible for the degradation of over 80% of the proteins in mammalian cells [130,131]. The post-translational protein modifications carried out by the UPS modulate multiple steps during NF-κB activation. In particular, in the canonical NF-κB pathway, IκB proteasomal degradation allows the release of active NF-κB (p50/RelA), while in the non-canonical pathway, NIK/IKKα triggers the partial proteolysis of phosphorylated p100 to form p52 [8]. Given this crucial role in NF-κB activation, the ubiquitin/proteasome pathway is a target for drug development [130]. The inhibition of the UPS can be carried out by targeting different UPS components such as the ubiquitin-activating enzymes (E1s), ubiquitin-conjugating enzymes (E2s), ubiquitin ligases (E3s), the 20S proteasome catalytic core (20S), and the 19S proteasome regulatory particles (19S) [131].

*Proteasome inhibitors*. Bortezomib (20S particle inhibitor) is the first proteasome inhibitor approved by FDA for the treatment of MM, MCL, WM, non-small-cell lung cancer (NSCLC), and pancreatic cancer [92]; additionally, it is now in clinical trials for DLBCL in combination with R-CHOP and for newly diagnosed or relapsed and refractory MM-in combination therapy with dexamethasone (NCT01965977; NCT03129828; NCT05052970; NCT04140162; NCT04717700; NCT05218603; NCT03896737; NCT03733691; NCT03110562). The results from these clinical investigations demonstrated that while the addition of bortezomib to R-CHOP did not improve the outcome of patients with non-germinal center B-cell-like (non-GCB)-DLBCL treated with R-CHOP alone [132,133], the combination therapy of bortezomib and dexamethasone was effective in increasing the progression-free survival (PFS) of relapsed and refractory MM patients previously treated with lenalidomide enrolled in a phase III study [134]. Furthermore, bortezomib monotherapy was shown to be well tolerated and effective for the management of MM patients after stem cell transplant [135,136]. Despite these encouraging results, the numerous side effects associated with bortezomib treatment, as well as the drug resistance onset, probably due to mutations in the proteasome subunit beta type-5 (PSMB5) [137], have limited its use for the treatment of solid tumors [138]. The second-generation proteasome inhibitor Carfilzomib (CFZ) is a selective anti-cancer drug targeting the chymotrypsin-like activity of the 20S proteasome [139]. CFZ has been approved by FDA as single agent or as part of combination regimens for treating relapsed and refractory MM patients who have already received other therapies [92,139]. Additional studies conducted in relapsed and refractory MM demonstrated that patients treated with CFZ, dexamethasone, and the anti-CD38 monoclonal antibody daratumumab showed an extended PFS as well as a favorable benefit risk profile compared to patients treated with CFZ and dexamethasone [140]. Additional clinical trials testing CFZ in MM or relapsed and refractory MM are still ongoing (NCT03934684; NCT02970747; NCT02199665; NCT02899052; NCT04813653; NCT04176718; NCT04263480; etc). To ameliorate drug administration, an orally bioavailable tripeptide analog of CFZ, Oprozomib, has been developed. Oprozomib showed anti-tumor activity in preclinical models and good efficacy in relapsed and refractory MM patients and those with WM when used as standalone or as combination therapy [141,142,143]. However, two recent phase Ib/II studies, OPZ003 (NCT01881789) and OPZ006 (NCT02072863), conducted in newly diagnosed MM, demonstrated that the use of Oprozomib in combination chemotherapy regimens exhibited gastrointestinal toxicities and variable pharmacokinetic (PK), suggesting that further optimization studies are needed [144]. To date, Ixazomib (MLN-9708), which binds to and blocks the 20S catalytic core of the proteasome, is the first and only orally available proteasome inhibitor approved by FDA for relapsed and refractory MM in combination with other antineoplastic drugs, as it was shown to be well tolerated and effective [145,146,147]. Furthermore, the use of Ixazomib as maintenance therapy extended the PFS of patients with newly diagnosed MM who did not undergo autologous stem cell transplantation; the drug was not associated with adverse effects [148]. Conversely, phase II trial of Ixazomib in relapsed and refractory systemic light-chain amyloidosis in combination with dexamethasone was discontinued due to the low improvement of patient outcomes [149]. Several clinical trials testing Ixazomib in different indications are still ongoing (NCT04837131; NCT03783416; NCT03618537; NCT03616782; NCT04047797; etc). The natural compound Marizomib (previously called NPI-0052) was approved as an orphan drug by the FDA for the treatment of MM patients, as it showed greater efficacy and less toxicity compared to other proteasome inhibitors [150,151,152]. Marizomib is also under investigation for patients with newly diagnosed GBM in combination with radiotherapy and chemotherapy (NCT03345095).

*Ubiquitin inhibitors.* Ubiquitin (Ub) and ubiquitin-like proteins (Ubl) control many physiological processes, including protein degradation, localization, and activation by promoting post-translational modifications. Given the important role of Ub and Ubl-dependent pathways, small drugs targeting E1-activating enzymes and E3 ligases have been developed [153]. PYR-41, a ubiquitin-activating enzyme E1 inhibitor, has been demonstrated to reduce inflammation via inhibition of NF-κB-mediated secretion of proinflammatory cytokines in the lungs of septic mice [154]. It was also shown to block angiotensin II-induced activation of dendritic cells in autoimmune diseases [155]. MLN4924 (Pevonedistat) was discovered to be a potent inhibitor of NEDD8 activating enzyme (NAE) which is able to induce apoptosis in vitro and suppress tumor growth in vivo. NAE is an important molecule that controls the turnover of several proteins by regulating the activity of cullin-RING-E3 ubiquitin ligases [156]. A phase Ib study of MLN4924 in combination with standard-of-care chemotherapies for the treatment of patients with solid tumors demonstrated that this therapeutic regimen was well tolerated, and no drug-related adverse events were reported [157]. Accordingly, MLN4924 is currently under investigation in a phase III clinical trial which is evaluating its efficacy and safety in combination with azacytidine, a cytotoxic chemotherapy drug, in AML patients who are not eligible for standard induction therapy (NCT04090736) [158]. Notably, other E1 inhibitors like TAK-243, TAS-4464 and ML-792 targeting ubiquitin-like modifier-activating enzyme 1 (UBA1), NAE and SUMO E1 activating enzyme, respectively, have been investigated in several cancer models, both in vitro and in vivo, and encouraging results have been reported [159,160,161,162]. These inhibitors exert their anti-tumor activity by triggering apoptotic cancer cell death, thus reducing tumor growth in xenograft mouse models [159,160,161,162]. Although TAS-4464 has shown successful preclinical results, the phase I study for the treatment of patients with solid and hematological cancers was discontinued due to severe liver toxicity (NCT02978235) [163]. A phase I trial investigating TAK-243 for the treatment of advanced solid tumors has terminated but no results have been reported yet, while its clinical evaluation for relapsed and refractory AML, refractory MDS, and CML (NCT03816319) is still ongoing. Therefore, to date, no E1 inhibitors have been approved.

Due to the high specificity of E3 ligases for their substrates, targeting these molecules has attracted a lot of interest in drug discovery, considering the reduced risk of off-target effects [131]. A lot of E3 inhibitors have been investigated, showing anti-cancer activity in preclinical models [164,165,166]. It is well known that the oncogene c-MYC is constitutively activated in MM, and its deregulation is mediated by E3 ligase HUWE1 [167,168]. Recently, Crawford and collaborators demonstrated that genetic and pharmacological inhibition of E3 ligase HUWE1 inhibits MM cell proliferation and induces MM cell cycle arrest with no effect on normal cells [169]. The authors observed a reduction of MYC expression in vitro, probably due to the deregulation of specific metabolic processes, including the reduction of intracellular glutamine levels. Furthermore, the combination of HUWE1 inhibitors with conventional chemotherapeutic drugs (i.e., proteasome inhibitor, lenalidomide) promoted synergistic anti-cancer activity in vitro. Likewise, HUWE1 genetic inhibition, in combination with chemotherapy, was shown to counteract tumor growth in vivo, suggesting that HUWE1 sustains tumorigenesis and that its deregulation would improve the efficacy of standard of care therapies [169]. These findings underline the possibility to therapeutically target HUWE1 in MM patients.

The NF-κB signaling pathway is activated by LUBAC, comprising the HOIL-1L, HOIP, and SHARPIN subunits [170]. Abnormal LUBAC activity was observed in several disorders, including ABC-DLBCL [171]. Recently, multiple LUBAC inhibitors (i.e., JTP-0819958 (HOIPIN-1) and its derivates) have been tested in vitro, where they showed inhibitory effects on the NF-κB pathway activation [172]. An in vitro assay identified Bendamustine as the best selective HOIP ligases inhibitor [173]. Bendamustine showed clinical efficacy with acceptable toxicity in relapsed TCL, NHL, B cell lymphoma, an untreated or relapsed CLL, both as monotherapy and in combination with other anti-cancer agents [174,175,176], suggesting that it could be used in standard of care regimens for treating these disorders.

*Deubiquitinating enzymes (DUBs) Inhibitors.* Deubiquitinating enzymes (DUBs) remove ubiquitin from target proteins [177]. Ubiquitylation processes control several physiological programs, and their deregulation contributes to the pathogenesis of many diseases [177]. For these reasons, DUB inhibitors have been investigated as potential anti-cancer agents in preclinical studies, although to date, none of them has reached clinical trials. In the last few years, using in silico assays, several potent and selective allosteric USP7 inhibitors have been identified (i.e., P50429, GNE-6640 and GNE-6776) [178]. Recently, a novel proteasome deubiquitinase inhibitor, VLX1570, was demonstrated to bind to and inhibit the activity of Ubiquitine Specific Proteases 14 (USP14) and ubiquitin C-terminal hydrolase L5 (UCHL5) in vitro. Moreover, VLX1570 treatment was shown to promote MM apoptosis and reduce MM growth in vivo. Although VLX1570 also showed anti-tumor effects in patients with relapsed and refractory MM, the phase I study was discontinued due to severe lung toxicity [179].

#### 2.2.3. NF-κB Transcription Factors

Selectively targeting NF-κB at the transcription factor level represents an attractive therapeutic strategy for the treatment of human hematological diseases. IT-901 is a novel inhibitor of the c-Rel and p65 NF-κB subunits. Treatment with IT-901 was shown to reduce lymphoma cell survival both in vitro and in vivo by blocking NF-κB-mediated oxidative stress response [180,181]. Accordingly, human primary CLL and Richter syndrome (RS) cells and CLL cell lines died by apoptosis upon IT-901 treatment, and no apparent toxicity was observed in normal B and T lymphocytes or in stromal cells [182]. IT-901 was also able to reduce tumor growth in vivo. Furthermore, reduced NF-κB binding to its consensus DNA site was observed in CLL cells as a consequence of NF-κB complex degradation in the cytosol. IT-901 induces anti-tumor effects by a dose-dependent increase in mitochondrial reactive oxygen species (ROS) along with a reduction of NF-κB-dependent transcription of genes encoding for Cytochrome c oxidase assembly subunit 2 (SCO2), ATP-synthase, and ATP5A1, as well as ROS scavenger, catalase (CAT), mitochondrial membrane potential, maximal Oxygen Consumption Rate (OCR), and ATP production [182]. The combination of IT-901 with ibrutinib, which is a first line therapy for CLL patients, increased the cytotoxic effect of both drugs, underling their potential use as combination therapy [182]. Collectively, this package of preclinical data showing the cancer-selective mode of action of IT-901 supports the further progression of this drug for clinical development.

Recently, the NF-κB transcription factor-PROTAC (NF-κB-PROTAC) was developed with the aim of selectively degrading p65 protein; it was proven to induce antiproliferative effects in cells, although in vivo studies are needed to validate both its efficacy and safety [121].

#### 2.2.4. NF-κB Nuclear Activities

The shuttling of NF-κB dimers between cytoplasm and nucleus is critical to promote NF-κB transcriptional programs [183]. The deregulation of this process has been observed in many disease conditions, suggesting that the inhibition of NF-κB nuclear activity (i.e., post-translational modification of NF-κB proteins and their ability to dimerize, translocate into the nucleus, bind to DNA, and interact with chromatin components, coactivators, and corepressors and other transcription factors) could be a potential therapeutic approach.

*Inhibitors of nuclear translocation*. NF-κB nuclear translocation is a highly controlled process which is involved different signals, i.e., nuclear localization signals (NLS), leucine-rich nuclear export signals (NES)), receptors (chromosome region maintenance 1/exportin1/Exp1/Xpo1 (CRM1)) and nuclear pore complexes (NPCs) [184,185]. Nuclear translocation of NF-κB is mediated by the NLS. After IκBα poly-ubiquitination and proteasomal degradation, NF-κB dimers interact with importin α/β and are carried into the nucleus through NPCs. The export of NF-κB is mediated by the CRM1-dependent pathway following interaction between the NES-IκBα-NF-κB complex and CRM1 [186]. Several new therapeutic candidates targeting NF-κB nuclear import and export are worthy of consideration. Leptomycin B (LMB) and its analogue KOS-2464 are irreversible CRM1 inhibitors that covalently bind to and inhibit CRM1/NES interaction, thus blocking p65 export [187,188]. Their use in clinical settings is limited due to the systemic toxicity and low efficacy observed in a phase I study [187]. A synthetic oral anti CRM1 agent, CBS9106, has been shown to block CRM1-dependent nuclear export via CMR1 protein degradation, leading to cancer cell apoptosis both in vitro and in vivo [189]. Several oral selective inhibitors of nuclear export (SINE) such as Selinexor (KPT-330) and Eltanexor (KPT-8602) have been developed and are under clinical evaluation for the treatment of many cancers as part of combination regimens (NCT03110562; NCT04854434; NCT05028348; NCT02649790). The results from a phase I clinical trial of oral Eltanexor in patients with relapsed and refractory MM demonstrated that this inhibitor was significantly efficient and well tolerated by MM patients; it achieved reduced expression of myeloma-dependent markers [190].

Although NF-κB nuclear import inhibitors have not entered clinical trials yet, many of them have been tested for their ability to inhibit NF-κB nuclear translocation, including synthetic small peptidomimetics SN-50, anti-inflammatory peptide-6 (AIP6), Ivermectin, Importazole, and dehydroxymethylepoxyquinomicin (DHMEQ) [191,192,193,194,195,196]. While Ivermectin and Importazole are importin α/β inhibitors, SN-50 blocks NF-κB nuclear import by inhibiting the NLS on the NF-κB complex [191,192,193,194]. AIP6 was demonstrated to directly interact with p65 and block the DNA-binding and transcriptional activities of the p65 NF-κB subunit in vitro. In addition, AIP6 displayed significant anti-inflammatory properties both in vitro and in vivo [195]. It has been demonstrated that DHMEQ promotes tumor regression and anti-inflammatory response in many cancer types including breast and ovarian and cisplatin-resistant NSCLC cancers, both in cell lines and xenograft mouse models [197,198,199,200]. DHMEQ exerts its anti-NF-κB activity by directly binding to NF-κB subunits (i.e., RelA, c-Rel or RelB) and blocking their nuclear translocation [196]. DHMEQ-mediated NF-κB inhibition was shown to induce downregulation of NF-κB-dependent anti-apoptotic genes and CLL cell death in vitro [197].

*Inhibitors of NF-κB transcriptional activity.* NF-κB transcriptional activity is also regulated by post-translational modifications such as ubiquitination, acetylation, methylation, SUMOylation, and phosphorylation [201,202]. Acetylation and deacetylation of NF-κB are under the control of histone acetyltransferase (HATs) and histone deacetylase (HDACs), respectively, and different coactivators (i.e., p300, CREB binding protein (CBP), HIV Tat-interacting protein 60 (Tip60)) [203]. The acetylation and methylation processes have been widely explored for therapeutic applications. Several HAT and HDAC, as well as DNA methyltransferase (DNTM) inhibitors (i.e., decitabine, vorinostat, romidepsin, belinostat, Panobinostat, tucidinostat and azacytidine), have been investigated in combination therapy in clinical settings, and most of them have been approved by FDA for various pathologies [65,204,205,206]. Among HDAC inhibitors, hydroxamic acid-based vorinostat (also known as SAHA and Zolinza) suppresses the activity of all HDACs except for HDAC III and exhibits anti-cancer functions including growth arrest promotion, activation of the extrinsic and/or intrinsic apoptotic pathways, induction of autophagic cell death, ROS-induced cell death, mitotic cell death, and senescence-selectively in tumor cells. The results of several clinical trials where vorinostat was investigated for the treatment of hematologic cancers showed that this drug was effective and well tolerated [207]. To date, two HDAC inhibitors, vorinostat and depsipetide, have been approved by the FDA to treat hematological malignancies such as CTCL [208,209]. Further investigations are needed to evaluate the use of these inhibitors as monotherapies for the treatment of solid cancers. Recently, a new agent targeting both PI3K and HDACs, namely CUDC-907, was investigated in preclinical tumor models, where the PI3K pathway and HDACs are constitutively activated. CUDC-907 demonstrated cytotoxic effects in prostate cancer cell lines in vitro and in a castration-resistant LuCaP 35CR patient-derived xenograft (PDX) model in vivo. It was shown that CUDC-907 induces DNA damage and apoptosis via c-Myc deregulation and suppresses tumor growth in vivo [127]. Several studies demonstrated that CUDS-907 synergized with BCL-2 inhibitor venetoclax to induce AML cell death in vitro and block AML growth in PDX models [210]. Likewise, CUDS-907 displayed more significant anti-tumor effects when used in combination with PARP inhibitor olaparib in SCLC cells and PDX models [211]. Accordingly, in a phase I trial of patients with relapsed/refractory DLBCL with MYC alteration, CUDS-907 demonstrated clinical safety and efficacy [212]. Additional clinical trials are ongoing to assess the efficacy of CUDC-907 in further oncological indications, including solid cancers.

*Inhibitors of NF-κB transactivation and DNA binding.* Another way to inhibit the NF-κB pathway is to target NF-κB transactivation, its binding to DNA, as well as the coactivators and corepressors involved in NF-κB-mediated transcription. Bromodomain-containing protein 4 (BRD4) belongs to the bromodomain and extra-terminal domain (BET) protein family, which is involved in the control of transcriptional machinery via binding to acetylated lysine residues of proteins, including NF-κB, [213,214]. OTX015 is the most characterized BET inhibitor. It is capable of disrupting the interaction between BET proteins and NF-κB dimers at promoters and has shown clinical efficacy in patients with refractory or relapsed hematological cancers [215,216]. Other BET inhibitors (i.e., cPI-0610, ZEN003694, BMS-986158) are being tested in several clinical trials for hematological and solid cancer, both as standalone treatments and in combination with other anti-cancer agents (NCT04986423; NCT05391022; NCT02158858; NCT03936465). A phase I study of GSK525762 (molibresib), a bromodomain and extra-terminal domain inhibitor, showed clinical benefit and acceptable adverse effects in patient with solid tumors, suggesting that this inhibitor could be a promising therapeutic agent [217]. New molecules have also been developed to overcome the limitation represented by the increased accumulation of BRD4 observed following treatment with conventional BET inhibitors. ARV-825 and ARV-771 PROTACs were shown to reduce BRD4 accumulation and induce more cancer cell apoptosis than BET inhibitors in hematological and solid cancers both in vitro and in vivo [218,219,220]. In addition, the treatment with these two BET-PROTACs reduced the expression levels of several pro-tumorigenic genes such as c-Myc, Cyclin-dependent kinase 4 (CDK4), cyclin D, B-cell lymphoma-extra large (Bcl-xL), XIAP, Cellular FLICE (FADD-like IL-1β-converting enzyme)-inhibitory protein (c-FLIP), c-IAP2, interleukin-10 (IL-10), and BTK—most of which are NF-κB-target genes—and, at the same time, increased the levels of anti-cancer genes like NOXA, p21 and p27, thus showing a higher efficacy than OTX015 [218,219,220].

NF-κB DNA binding inhibitors such as parthenolide (PN, a sesquiterpene lactone (SL)) and dimethylaminoparthenolide (DMAPT) are potent anti-inflammatory compounds which are able to covalently bind to the cysteine-38 residue of the p65 NF-κB subunit and inhibit p65 DNA binding, thus deregulating NF-κB activation [221]. To date, several preclinical studies of parthenolide in combination with other drugs such as DMAPT, HDAC inhibitors, and anthracyclines have been conducted; they showed that PN increases cancer cell apoptosis and reduces drug resistance, despite the poor bioavailability of this compound [222,223]. Conversely, DMAPT has a better bioavailability than PN; it showed anti-tumor activity both in vitro and in vivo in many cancer types and is thus considered promising for clinical translation [224,225,226].

Targeted drug delivery to tumor sites could be a potential new strategy to ameliorate internalization and avoid off-target toxicity. Emerging evidence showed that the use of aptamers, proteins, and antibodies conjugated to anti-tumor drugs and specifically targeting with high affinity molecules which are overexpressed in cancer cells is an active area of interest [227]. Recently, synthetic double-stranded oligonucleotides (ODNs) targeting NF-κB transcription factor, also known as NF-κB decoy ODNs, were conjugated to the RNA aptamer against transferrin receptor (TfR) and Doxorubicin (Dox) to generate an ODN chimera, named aptacoy. Aptocoy is able to selectively target pancreatic tumor cells by recognizing TfR—which is overexpressed in pancreatic cancer—and releasing Doxorubicin along with NF-κB decoy ODNs to sensitize tumor cells to Dox-induced cell death [228]. Accordingly, pancreatic cancer cells treated with aptacoy showed a reduction of NF-κB target gene expression and increased apoptosis in vitro [228]. Despite promising preclinical data, aptacoy has not entered clinical trials yet due to its poor uptake into cells, and further investigations are needed to optimize the physicochemical and pharmacological properties of these molecules.

Despite the clinical improvements observed following treatment with core pathway inhibitors in specific tumors, concerning proteasome inhibitors in MM, these anti-cancer drugs target the NF-κB pathway in both cancer and normal cells, often resulting in severe toxic effects at effective dose levels. In addition, drug resistance remains a major obstacle to the successful treatment of cancer, suggesting that further investigations are needed to overcome these limitations. In this regard, new drug candidates have progressed to phase III trials and seem to be well tolerated (i.e., Icaritin and MLN4924), while others (i.e., IT-901) showed cancer-selectivity in preclinical studies. Furthermore, novel therapeutic tools like PROTACs, which provide a platform to achieve selective degradation of specific NF-κB pathway components, are being tested in vitro. While these new therapeutics hold promise, they need to be clinically validated, underscoring that the development of disease-specific, clinically useful anti-NF-κB agents is still an unmet medical need.

### 2.3. Inhibitors of NF-κB Downstream Effectors

The barrier to therapeutically targeting NF-κB has been achieving disease-cell specificity, given the ubiquitous functions of NF-κB. Since NF-κB drives pathology by inducing tissue- and context-specific transcriptional programs, several research groups have pursued an alternative approach to overcome this barrier, i.e., targeting essential, specific effectors of NF-κB pathogenetic functions, including the promotion of cell survival, angiogenesis, inflammation, and metabolic adaptation, which are fundamental processes in a range of human disorders, including cancer [20,229] (Figure 5) (Table 1).

In MM, an NF-κB-driven cancer of plasma cells (PCs), our group identified the complex formed by growth arrest and DNA damage inducible beta (GADD45B) and the JNK kinase, mitogen-activated protein kinase kinase 7 (MKK7), as an essential survival module downstream of NF-κB [230,231] and a therapeutic target in MM [229]. GADD45B belongs to the GADD45 family of proteins, along with GADD45α and GADD45γ, which are involved in several key processes such as DNA repair and demethylation, cell-cycle regulation, senescence, and apoptosis [232,233]. We developed a first-in-class GADD45B/MKK7 inhibitor, DTP3, which selectively kills MM cells via MKK7/JNK-driven apoptosis and is not toxic to normal cells. Owing to this cancer-cell specificity, DTP3 ablated MM xenografts in mice, extending overall survival (OS) in orthotopic models without adverse effects. Due to its mode of action, i.e., operating downstream of NF-κB, DTP3 bypassed resistance to multiple drugs used in MM (i.e., steroids, immunomodulatory drugs, proteasome inhibitors), creating a unique profile of clinical utility for development in MM and DLBCL [229,234,235]. Regulatory studies confirmed the on-target pharmacology, anti-cancer efficacy, and favorable PK and ADME profiles of DTP3, coupled with tolerability, leading to the first-in-human pilot trial in MM patients. The results from this study confirmed the clinical capacity of DTP3 to trigger JNK activation and apoptosis in MM but not in normal cells whilst producing clinical benefit according to the IMWG criteria with no significant adverse events [234]. Notably, DTP3 produced these pharmacodynamic (PD) and initial efficacy signals in refractory MM patients. As such further investigation of its novel mode of action in a larger follow-on trial is warranted [234,235]. 

Among NF-κB anti-apoptotic target genes are B-Cell Lymphoma 2 (BCL-2) family members including Bcl-X_L_, myeloid cell leukemia sequence 1 (MCL1), and A1 and B-cell lymphoma 2 (BCL-2). These proteins contribute to the onset of NF-κB-dependent drug resistance and are overexpressed in several cancer types such as breast and prostate cancers and B-cell malignancies [10,236]. Accordingly, the approach of targeting BCL-2 proteins has been pursued for cancer treatment. Over the last ten years, “BH3-mimetics” drugs—that mimic the pro-apoptotic functions of the BH3 domains of Bcl-2 proteins—have been the principal approach to blocking the anti-apoptotic function of Bcl-2 family members [237]. Although several agents have been tested in preclinical and clinical studies, only ABT-199 (venetoclax) has been approved by the FDA for the treatment of patients with (1) CLL; (2) SLL patients; and (3) newly diagnosed AML-aged 75 years or older, or who have comorbidities precluding intensive induction chemotherapy, in combination with azacitidine or decitabine or low-dose cytarabine (LDAC) [10,236,238,239,240]. These BH3-mimetics also showed clinical efficacy in other hematological malignancies like FL, MCL, and MM, both alone and in combination with other drugs [239,241]. In addition, Venetoclax is currently being tested in combination therapies for the treatment of acute lymphoblastic leukemia (ALL) and DLBCL [239]. Moreover, several other BCL-2, BCL-XL and MCL-1 inhibitors (i.e., GX15-070 (obatoclax), AT-101, BI-97C1 (sabutoclax) ABT-737 (navitoclax)) are under investigation as single agents or in combination therapies in hematological and solid cancers to improve the selectivity for a specific Bcl-2 member and the safety profile [236,242]. In addition to BH3 mimetics, the use of BH3 peptides, G-quadruplex (G4) motif, as well as the targeting of the BH4 domain, could represent additional strategies to inhibit Bcl-2 members, as shown in recent preclinical studies [236,243,244].

The principal effectors of NF-κB pro-angiogenetic function belong to the vascular endothelial growth factor (VEGF) family, which comprises five members: VEGF-A, VEGF-B, VEGF-C, VEGF-D, and placenta growth factor (PIGF). These factors interact with several tyrosine-kinase receptors such as VEGFR-1, VEGFR-2, and VEGFR-3 to promote tumor-associated angiogenesis, tissue infiltration, and metastasis formation [245,246]. Bevacizumab (BVZ) is a humanized anti-VEGF-A monoclonal antibody which is able to block the interaction between VEGF-A and its receptor by binding circulating and soluble VEGF-A [246]. BVZ is the first anti-angiogenic drug approved by FDA and EMEA for the treatment, in combination with chemotherapy, of metastatic CRC, OC, breast, renal, and NSCLC cancers, and as monotherapy for the treatment of GBM. The combination of BVZ with immune checkpoint inhibitors is currently under clinical investigation and could represent a promising treatment option for patients with unresectable HCC or untreated metastatic renal cell carcinoma (mRCC) (NCT03434379; NCT02715531; NCT02420821) [247,248]. Along with BVZ, Aflibercept was approved by the FDA in combination with chemotherapy for the treatment of metastatic CRC [92]. Moreover, the VEGFR-1 inhibitor, Sunitinib, has also been approved by FDA for the treatment of renal cell cancer, gastrointestinal stromal tumors (GISTs), and progressive, well-differentiated pancreatic neuroendocrine tumors (pNETs) [92,249].

Additional VEGFR inhibitors like brivanib alaninate (BMS-582664), motesanib (AMG 706), and linifanib (ABT 869) have been clinically investigated and showed good efficacy, while vandetanib (ZD6474), axitinib (AG-013736), and cediranib (AZD2171) are still under evaluation in ongoing clinical trials (NCT00537095; NCT00920816; NCT00942877) [250,251]. Collectively, these data support the use of angiogenetic inhibitors in combination with standard-of care chemotherapy for the treatment of several incurable advanced tumors.

The role of NF-κB in mediating tumor-promoting inflammation has been widely recognized [18]. Proinflammatory cytokine interleukin-6 (IL-6) is one of the key proinflammatory genes induced by NF-κB; its role in tumorigenesis and inflammatory diseases has been extensively elucidated. Effective anti-IL-6 agents have been developed and, to date, the monoclonal antibodies (mAb) tocilizumab (anti-IL-6R) and siltuximab (anti-IL-6) are approved by the FDA for the treatment of RA and Castleman’s disease, respectively [252]. The use of IL-6 blockers as anti-cancer agents has been evaluated in many cancers (i.e., lung, prostate cancers, OC, B-cell NHLS, renal cell carcinoma) [252]. Although Elsilimomab (BE-8) did not show clinical efficacy in cancer patients, other anti-IL-6 mAbs such as mAb 1339 (OP-R003) and ALD518/BMS-945429 have exhibited potential anti-tumor activity in vivo, inducing significant clinical response in patients with different disorders [252], suggesting that they could be used to treat cancer. Furthermore, the anti-IL-6 mAb, siltuximab, is being tested in clinical trials for several cancers such as large granular lymphocytic leukemia (LGLL), metastatic pancreatin cancer, and MM, both alone and in combination therapy (NCT05316116; NCT04191421; NCT03315026). Recently, Hailemichael and collaborators showed that IL-6R blockade abrogates immunotherapy-associated toxicity, thus reinforcing anti-tumor immunity in vivo [253]. Accordingly, the combination therapy of tocilizumab and immune checkpoint inhibitors (anti-PD-1 and anti-CTLA-4) is currently in phase II trial for patients with melanoma, NSCLC, and urothelial carcinoma (NCT04940299) [56,254].

In recent decade, a growing body of evidence has underscored the key role of NF-κB signaling in governing metabolic adaptations to environmental changes and the disruptions of tissue homeostasis that occur in disease [255]. Recently, several upstream activator and downstream effectors mediating NF-κB-dependent metabolic rewiring have been identified and represent promising targets, including c-Myc, CCAAT enhancer binding protein beta (CEBPB), glutamate dehydrogenase 1 (GDH1), 6-phosphofructo-2-kinase/fructose-2,6-biphosphatase 3 (PFKFB3), and carboxylesterase 1 (CES1). While some of them, like c-Myc, are still considerable undruggable [256], others, like PFKFB3 and CES1, were reported to be actionable targets in different preclinical cancer models. It was shown that under glutamine deprivation, IKKβ promotes cell survival by phosphorylating and inhibiting PFKFB3, one of four tissue-specific PFKFB isoenzymes identified so far which is involved in glycolysis. Consistently, the pharmacological co-inhibition of IKKβ and glutamine metabolism resulted in the synergistic killing of cancer cells, both in vitro and in vivo [257]. Recently, CES1 has been identified as an essential NF-κB-regulated lipase linking obesity-associated inflammation with metabolic adaptation to energy stress in aggressive CRC [258,259,260]. CES1 expression was upregulated in consensus molecular subtypes (CMS)4 and CMS2 tumors and correlated with worse clinical outcomes in overweight CRC patients [258,259,260], suggesting a role for CES1 in the pathogenesis of CRC. Consistent with this idea, treatment with CES1 inhibitors effectively killed human CRC cells upon glucose limitation and markedly impaired CRC growth without apparent adverse effects in mouse allograft and xenograft models [258,259,260]. Collectively, these results identify CES1 as a promising therapeutic target, given its contextual specificity for energy stress conditions, correlation with poor clinical outcomes in obese CRC patients, and clear stratification with MSS/non-hypermutated CMS4 and CMS2 tumors, which do not respond to immunotherapy [258,259,260]. Accordingly, genetic CES1 deletion appears to be tolerated in vivo, since CES1 knockout mice are viable, lean, and die of old age [261].

Collectively, this bulk of evidence demonstrates that the safe and cancer-selective inhibition of the NF-κB pathway is clinically achievable and promises profound benefit to patients with NF-κB-driven cancers.

## 3. Conclusions

Transcription factor NF-κB drives oncogenesis. Aberrant NF-κB activation is a hallmark of several human malignant and non-malignant diseases. Despite the massive effort by the pharmaceutical industry to develop a specific NF-κB inhibitor, none has been clinically approved due to the dose-limiting toxicities associated with the global suppression of NF-κB (i.e., IKK inhibitors). Recently, novel subunit-specific inhibitors such as the anti-c-Rel, IT-901, have been developed and seem to be well tolerated in mice. However, the clinical safety of these new agents remains to be demonstrated. The strategy of targeting specific essential upstream activators/downstream effectors of the pathological functions of NF-κΒ seems to be an alternative, safer route to circumvent the limitations of conventional IKK/NF-κB-targeting drugs and to overcome drug resistance. Despite the fact that these therapeutic approaches are not NF-κB specific, they might be clinically useful nonetheless, in particular if the pathways (i.e., mTOR, AKT) targeted beyond NF-κΒ cooperate with it in driving human diseases. Therefore, the next step would be to thoroughly characterize the transcriptional programs elicited by NF-κΒ as well as the genetic drivers responsible for the constitutive activation of this pathway in pathologic conditions to identify actionable therapeutic targets that may lead to better diagnostics and therapeutics for human diseases.

## Figures and Tables

**Figure 1 biomedicines-10-02233-f001:**
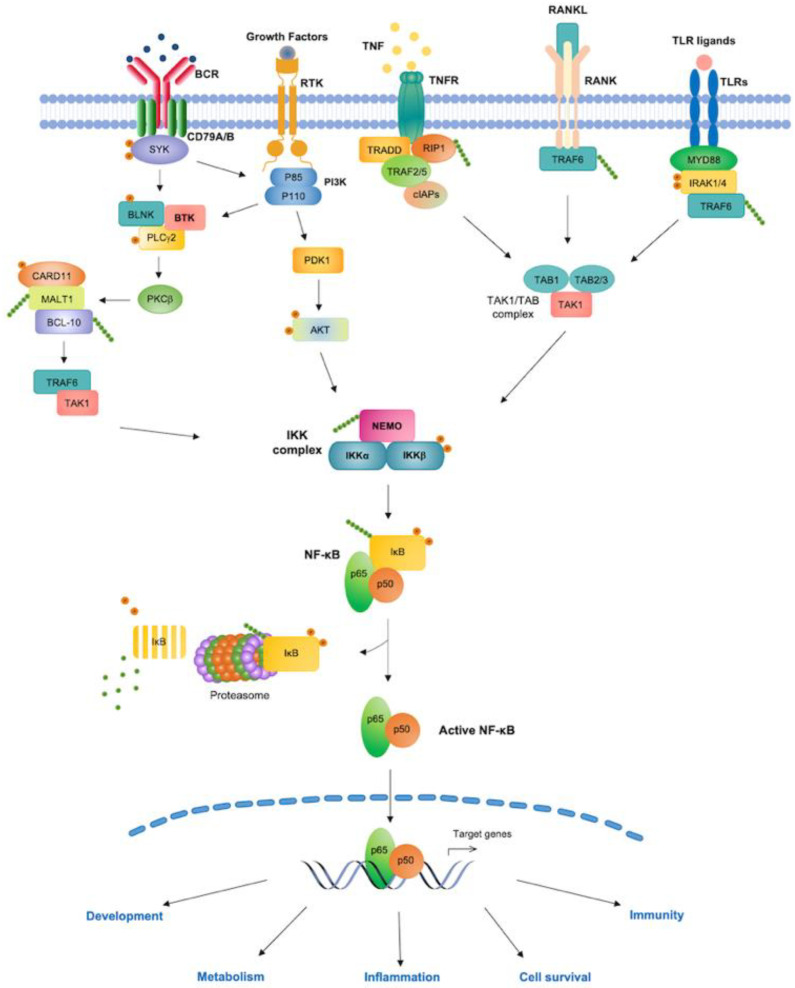
Schematic representation of the canonical NF-κB pathway. The exposure to activating stimuli triggers the activation of receptor-specific transduction pathways converging on the recruitment of IKK complex via NEMO, which, in turn, allows the TAK1 kinase to activate IKKβ. IKK complex activation leads to IκBs’ phosphorylation and degradation via ubiquitin-proteasome system, and the following release of active NF-κB dimers, which translocate to the nucleus to activate their target genes.

**Figure 2 biomedicines-10-02233-f002:**
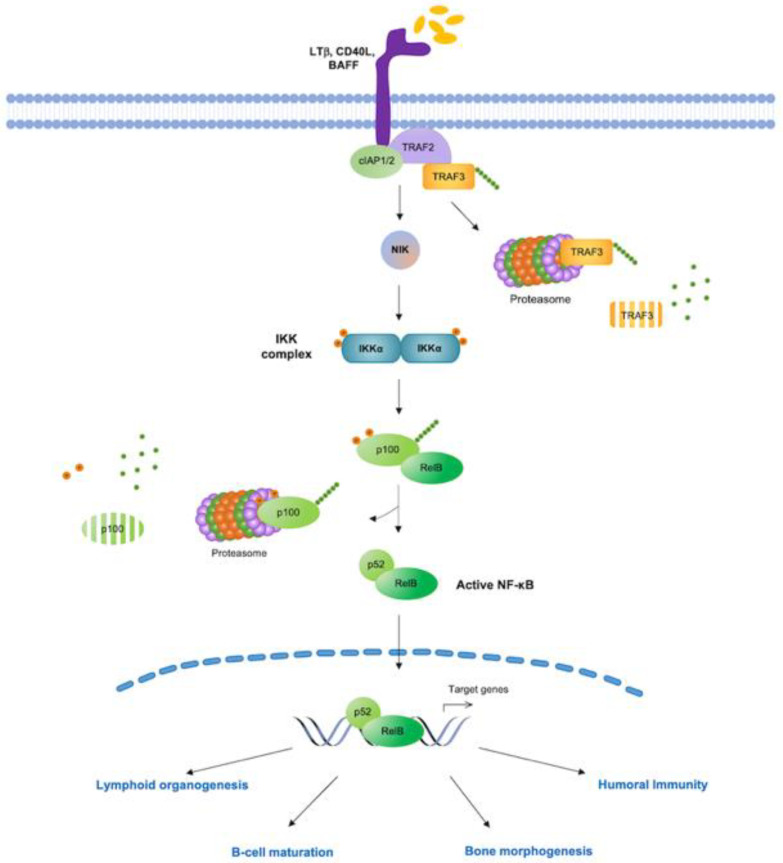
Schematic representation of the non-canonical NF-κB pathway. The activating event of this pathway is represented by the signal-induced stabilization of NIK and subsequent NIK-mediated phosphorylation of IKKα. In the absence of the proper stimulation, NIK is constitutively degraded by proteasome following E3 ubiquitin ligase c-IAP1/2-mediated ubiquitination. Upon ligand-receptor binding, TRAF3 is degraded resulting in NIK stabilization and phosphorylation of IKKα. In turn, this event leads to the proteasome-mediated processing of p100 to p52 and the translocation of p52/RelB heterodimers to the nucleus, where they activate their transcriptional programs.

**Figure 3 biomedicines-10-02233-f003:**
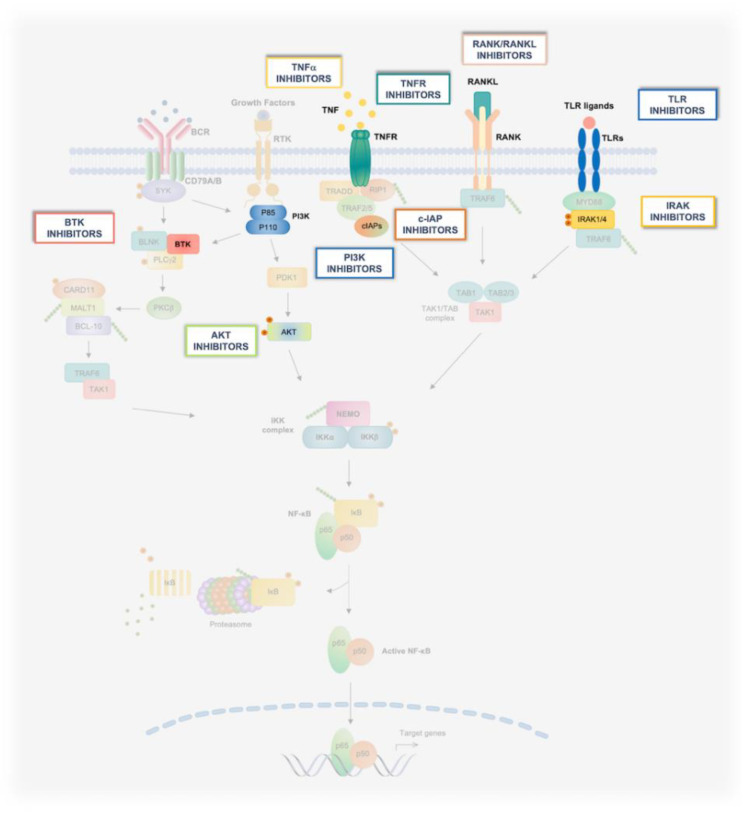
NF-κB inhibitors acting by blocking activators upstream of the IKK complex in the canonical NF-κB pathway.

**Figure 4 biomedicines-10-02233-f004:**
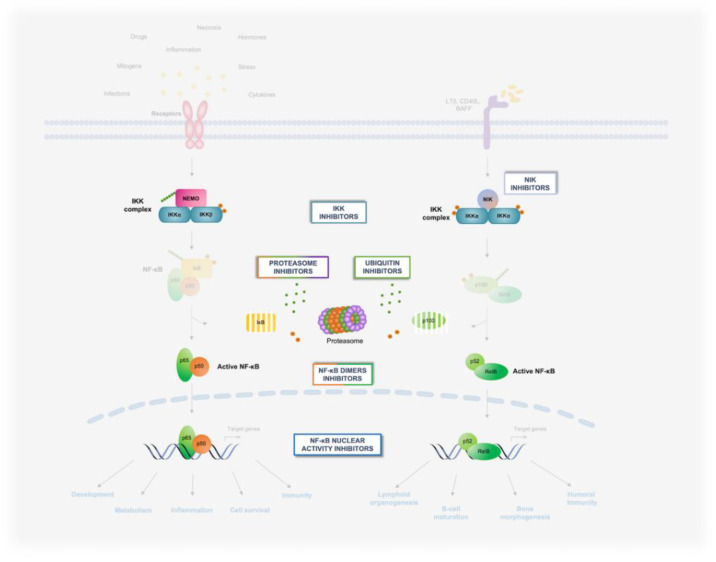
NF-κB-targeted therapeutics acting by inhibiting the canonical and non-canonical NF-κB core pathway.

**Figure 5 biomedicines-10-02233-f005:**
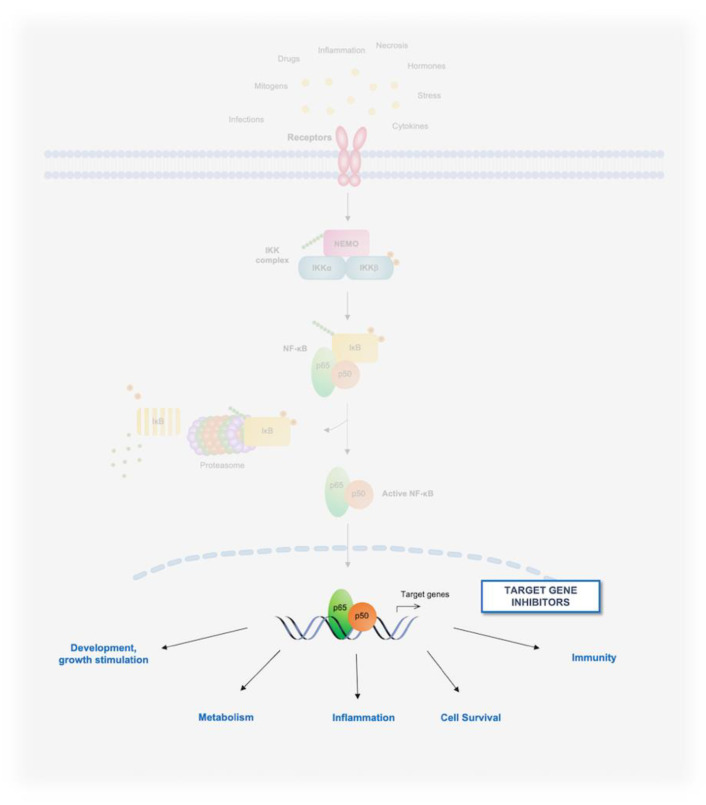
NF-κB inhibitors acting by blocking downstream effectors of canonical NF-κB pathogenetic functions.

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
