# Peer review of "The NF-κB Pharmacopeia: Novel Strategies to Subdue an Intractable Target"

_biomedicines, 2022, doi:10.3390/biomedicines10092233_

Round 1

Reviewer 1 Report

NF-kB is a widely studied and widely commented on pathway and transcription factor, in part due to its implication in many aspects of tumor biology. As the authors describe, there have been many efforts to develop NF-kB pathway inhibitors for cancer therapy, but many of these have been abandoned due to more general toxicity. Thus, the authors, for the most part, describe situations where upstream or downstream inhibition of NF-kB has been attempted, and the success or failures of such efforts.

As such, this article provides a relevant and valuable compendium of alternatively methods to target “NF-kB” signaling. Nevertheless, I would argue that some of the discussions no doubt involve non-NF-kB pathways for activation, and thus targeting.

In spite of the title and much hype throughout the article, the concluding remarks section iis underwhelming in that they do not really describe what might be actually valuable targets. Indeed, throughout the article, the authors describe somewhat ‘alternative’ methods for targeting NF-kB but never really point to highly promising or successful approaches, targets, and cancers for NF-kB ‘therapy’. I see this as the major limitation of the review, the authors do not really distinguish any of the “additional” approaches as ones that might prove “most promising”. Rather, they end basically concluding that NF-kB targeting may be insurmountable, which is not what the title suggests they will demonstrate.

Based on the number of “typos” that I discovered (below), the authors are advised to do another thorough read-through of this article.

Comments

--Abstract, maybe soften “large majority of human cancers”, not sure what the large majority is (60%, 95%, etc.)…..”many human cancers” should be sufficient there

--line 37, not sure “motif” is the current word for referring to the RHD….motif implies a much small amino acid sequence than something that is about 300 aa’s long

--line 74, I think the word “of” is missing

--Line 106, probably remove hyphen in “cancer-cell”

--There is much writing that is a bit over-exuberant.  For example, highlighted in italics in this section:  

This vast existing body of genetic, biochemical, and clinical evidence provides a compelling rationale for therapeutically targeting the NF-κB pathway in a wide range of hu- man cancers. However, despite the aggressive efforts made by the pharmaceutical industry over the last three decades, no specific NF-κB inhibitor has been clinically approved, due to the dose-limiting, on-target toxicities of systemic NF-κB blockade [10,20,24]. Underscoring this effort, in 2006 – just before IKK inhibitors could be tested in animal models and human trials – there were already no fewer than 750 drug candidates aimed at blocking pathologic NF-κB activation [3,24]. However, notwithstanding these tremendous efforts and the enormous progress made ever since in unravelling the complex molecular mechanisms regulating the pathways of NF-κB activation and their biological functions in health and disease, the output of actionable medicinal interventions for targeting the IKK/NF-κB system in the clinical setting has been dismally scant.

--Figure 3, I am guessing that many of these upstream inhibitors target pathways in addition to NF-kB.  Some comment on that necessary.

--Section 2.1,2. Line 35.  Is it clear why the TLR drug IMO-8400 works in MYD88 mutant cancers?  Shouldn’t NF-kB activation be occurring downstream of the drug’s target?

---In same section, are the IRAK inhibitors part of NF-kB inhibition? Unclear

--Section 2.2, line 235, “develop” should be “development”.  Line 236, insert “the” before NF-kB core pathway.

---"Despite the goal has been to develop specific IKKβ inhibitors,” awkward phrasing (“Although the goal’?)

--Line 255, do they mean “thio” reactive?

-sentence before, maybe don’t use “MoA” abbreviation

---"NF-κB- mediated” line 307, close hyphenated words

--line 317, insert space before reference “in clinical trials[90].”

--Line 323, insert “an”, “generation of ‘an’ analogue series”

--line 328, fix “In human Human peripheral blood…”

--Line 332 “An emerge, powerful class of…”  use “emerging”

--Line 460, page 10, Insert “The” at beginning of sentence/paragraph

--page 22, line 504, in vivo in italics as elsewhere in paper

--page 24, line 612, I think should be “deregulating”

--Line 723, I think it should be “trials” plural

---Lines 736, 742, 758, 770 italics on in vivo in all places

--Lines 776 and 781, “kappa” on NF-kB

--In Conclusion “Despite the massive effort by the pharmaceutical industry to develop a specific NF-κB inhibitor, none has been clinically approved,” Is that really true?

Reviewer 2 Report

This is very interesting and well-written paper with 40 pages. This reviewer thinks that this paper covers a lot of points in NF-kB pathways and their inhibitors. Since this paper has too many things, which makes readers less focused, rather, some unnessary parts or weakly importnat parts can be deleted or shorten. For examples, IMO-8400 parts, HDAC and DNMT inhibitors can be deleted. 

Some typo errors (eg., L294: TNF-alpha, L491, L504: in vivo, L776: NF-kappaB, and others) should be amended.

Chemical structures of some representative inhibitors should be presented.

Title of references should be ameded to lowercase letter.

REF #34, 37, 42, 54, 66, 75, 96, 100, 106, 113, 122, 149, 150, 190, 199, 208, 223/238: correct Journal name !!

Reviewer 3 Report

The Review article Titled  "The NF-κB Pharmacopeia: Novel Strategies to Subdue an Intractable Target" Authored Daniela Verzella and colleagues provides a wealth of data regarding pharmacautical targeting of NF-κΒ pathway as a means to fight disease and in partocular neoplasia.

My Comments and Suggestions for Authors are the following:

1) the review is well-written and easy to follow. There are few errors (for example line 79 noncanonical -> non-canonical) that the authors can easily detect and revise following a thorough review.

2) NF-kB pathway is a well-studied topic with many articles and reviews. Hovewer the approach of the authors is interesting, the bibliography well-studied and presented and the conclusions comprehensively described. I would recommend that the authors might tone-down the word "most" in line 777 of conclusions, or to separate the sentence into malignant and non-malignant disease. 

Reviewer 4 Report

This review is very well written. It contains necessary and sufficient information and supplies useful information to the reader.

Round 2

Reviewer 1 Report

Point 3.  i would still argue that the RHD is not an amino acid motif.  It is a

larger complex domain with several motifs within it (for DNA binding,

dimerization, NLS, etc)......why not call it an "amino-acid region"
